# Novel *DNM1L* variants impair mitochondrial dynamics through divergent mechanisms

Kelsey A Nolden[1],*, John M Egner[1],*, Jack J Collier[2,3], Oliver M Russell[2], Charlotte L Alston[2,4], Megan C Harwig[1], Michael E Widlansky[5], Souphatta Sasorith[6], Inês A Barbosa[7], Andrew GL Douglas[8,9], Julia Baptista[10], Mark Walker[11], Deirdre E Donnelly[12], Andrew A Morris[13], Hui Jeen Tan[14], Manju A Kurian[15], Kathleen Gorman[16,17], Santosh Mordekar[18], Charu Deshpande[19], Rajib Samanta[20], Robert McFarland[2,4], R Blake Hill[1], Robert W Taylor[2,4], Monika Oláhová[2]

Imbalances in mitochondrial and peroxisomal dynamics are associated with a spectrum of human neurological disorders. Mitochondrial and peroxisomal fission both involve dynamin-related protein 1 (DRP1) oligomerisation and membrane constriction, although the precise biophysical mechanisms by which distinct DRP1 variants affect the assembly and activity of different DRP1 domains remains largely unexplored. We analysed four unreported de novo heterozygous variants in the dynamin-1-like gene *DNM1L*, affecting different highly conserved DRP1 domains, leading to developmental delay, seizures, hypotonia, and/or rare cardiac complications in infancy. Single-nucleotide DRP1 stalk domain variants were found to correlate with more severe clinical phenotypes, with in vitro recombinant human DRP1 mutants demonstrating greater impairments in protein oligomerisation, DRP1-peroxisomal recruitment, and both mitochondrial and peroxisomal hyperfusion compared to GTPase or GTPase-effector domain variants. Importantly, we identified a novel mechanism of pathogenesis, where a p.Arg710Gly variant uncouples DRP1 assembly from assembly-stimulated GTP hydrolysis, providing mechanistic insight into how assembly-state information is transmitted to the GTPase domain. Together, these data reveal that discrete, pathological *DNM1L* variants impair mitochondrial network maintenance by divergent mechanisms.

## Introduction

In response to various environmental and cellular stimuli, the mitochondrial network undergoes continuous architectural remodelling. The morphology of the mitochondrial network is controlled by two dynamic events—mitochondrial fission and fusion (Kasahara & Scorrano, 2014; Mishra & Chan, 2014; Dorn et al, 2015; Roy et al, 2015; Touvier et al, 2015; Wai & Langer, 2016; Harvey, 2019). The balance of these events is essential for even distribution of mitochondrial content, mitochondrial protein quality control, and regulation of mitochondrial activity. Besides regulating mitochondrial metabolism, mitochondrial fission and fusion events play an essential role in a number of cellular processes, including cell cycle regulation (Qian et al, 2012; Horbay & Bilyy, 2016; Pangou & Sumara, 2021), immune response (Cervantes-Silva et al, 2021), and cell death (Aouacheria et al, 2017).

Mitochondrial fusion is largely mediated by the outer mitochondrial membrane proteins mitofusin 1 (MFN1) and mitofusin 2 (MFN2) and the inner mitochondrial membrane protein optic atrophy 1 (OPA1). Perturbed mitochondrial fusion leads to morphological

[1]Department of Biochemistry, Medical College of Wisconsin, Milwaukee, WI, USA   [2]Wellcome Centre for Mitochondrial Research, Newcastle University, Translational and Clinical Research Institute, Faculty of Medical Sciences, Newcastle upon Tyne, UK   [3]Department of Neurology and Neurosurgery, Montreal Neurological Institute, McGill University, Montreal, Canada   [4]The National Health Service (NHS) Highly Specialised Service for Rare Mitochondrial Disorders, Newcastle upon Tyne Hospitals NHS Foundation Trust, Newcastle upon Tyne, UK   [5]Department of Medicine, Division of Cardiovascular Medicine and Department of Pharmacology, Medical College of Wisconsin, Milwaukee, WI, USA   [6]Laboratoire de Génétique Moléculaire, Centre Hospitalier Universitaire and PhyMedExp, INSERM U1046, CNRS UMR 9214, Montpellier, France   [7]Department of Medical and Molecular Genetics, School of Basic and Medical Biosciences, King's College London, London, UK   [8]Wessex Clinical Genetics Service, University Hospital Southampton NHS Foundation Trust, Southampton, UK   [9]Human Development and Health, Faculty of Medicine, University of Southampton, Southampton, UK   [10]Peninsula Medical School, Faculty of Health, University of Plymouth, Plymouth, UK   [11]Department of Cellular Pathology, University Hospital Southampton NHS Foundation Trust, Southampton, UK   [12]Northern Ireland Regional Genetics Centre, Belfast Health and Social Care Trust, Belfast City Hospital, Belfast, UK   [13]Willink Metabolic Unit, Manchester Centre for Genomic Medicine, Manchester University Hospitals NHS Foundation Trust, Manchester, UK   [14]Department of Paediatric Neurology, Royal Manchester Children's Hospital, Manchester University Hospitals NHS Foundation Trust, Manchester, UK   [15]Developmental Neurosciences Department, Zayed Centre for Research into Rare Diseases in Children, University College London Great Ormond Street Institute of Child Health, Faculty of Population Health Sciences, London, UK   [16]Department of Neurology and Clinical Neurophysiology, Children's Health Ireland at Temple Street, Dublin, Ireland   [17]School of Medicine and Medical Science, University College Dublin, Dublin, Ireland   [18]Department of Paediatric Neurology, Sheffield Children's Hospital, Sheffield, UK   [19]Clinical Genetics Unit, Guys and St. Thomas' NHS Foundation Trust, London, UK   [20]Department of Paediatric Neurology, University Hospitals Leicester NHS Trust, Leicester, UK

Correspondence: monika.olahova@ncl.ac.uk
*Kelsey A Nolden and John M Egner contributed equally to this work.

changes characterised by the presence of fragmented mitochondria. Conversely, mitochondrial fission leads to the division of mitochondria and impairment of this process causes the formation of hyperfused mitochondrial networks (Tilokani et al, 2018; Dorn, 2019; Collier & Taylor, 2021).

The GTPase dynamin-1-like protein (also referred to as Dynamin Related Protein 1 or DRP1), encoded by the *DNM1L* gene, is the central effector of mitochondrial division. DRP1 is predominantly found in the cytosol, but upon activation is recruited to the outer mitochondrial surface by membrane anchored receptor proteins—including mitochondrial fission factor (MFF), mitochondrial fission protein 1 (FIS1), and the mitochondrial dynamics proteins (MID49 and MID51) (Smirnova et al, 2001; James et al, 2003; Yoon et al, 2003; Stojanovski et al, 2004; Gandre-Babbe & Van Der Bliek, 2008; Otera et al, 2010; Palmer et al, 2011; Liu et al, 2013; Losón et al, 2013; Ihenacho et al, 2021)—to mediate mitochondrial fission. DRP1 assembles at mitochondria–ER contact sites (Friedman et al, 2011), organising into higher order oligomeric complexes that encompass mitochondrial tubules in a circumferential manner in either a helical (Mears et al, 2011; Fröhlich et al, 2013; Kalia et al, 2018) or filamentous organisation (Kalia et al, 2018). Subsequent GTP binding and hydrolysis drives conformational changes in oligomeric DRP1 structures, resulting in constriction of the membrane diameter, before a concert of interactions between mitochondria, other organelles, and vesicles trigger scission (Mears et al, 2011; Koirala et al, 2013; Basu et al, 2017; Kraus & Ryan, 2017; Kalia et al, 2018; Nagashima et al, 2020). Peroxisomal fission is independent of mitochondrial fission but requires many components of the mitochondrial fission apparatus, including DRP1, MFF, and FIS1 (Li & Gould, 2003; Koch et al, 2005; Kobayashi et al, 2007; Gandre-Babbe & Van Der Bliek, 2008; Otera et al, 2010; Koch & Brocard, 2012; Yamano et al, 2014).

The importance of mitochondrial division and dynamics is evidenced by the fact that *Dnm1l*⁻/⁻ knockout mice are embryonic lethal (Ishihara et al, 2009; Wakabayashi et al, 2009). Furthermore, cardiac-specific (Ashrafian et al, 2010; Ikeda et al, 2015; Ishihara et al, 2015; Song et al, 2015) and brain-specific (Ishihara et al, 2009; Wakabayashi et al, 2009) ablation of DRP1 leads to lethal dilated cardiomyopathy and defective cerebellar development with early postnatal death, respectively. Defects in human mitochondrial dynamics caused by de novo monoallelic or biallelic pathogenic *DNM1L* variants are often associated with developmental delay, hypotonia and neurological disorders, including encephalopathy, refractory seizures, and/or autosomal dominant optic atrophy (Table S1). It has been suggested that de novo heterozygous *DNM1L* variants likely exert a dominant-negative effect over the wild-type allele, impairing its ability to effectively achieve mitochondrial division (Whitley et al, 2018). However, the biophysical basis of impaired mitochondrial dynamics underpinned by human *DNM1L* variants remains unresolved. The first reported pathogenic *DNM1L* (NM_012062.5) variant, c.1184C>A, p.Ala395Asp (Waterham et al, 2007), located in the stalk domain of DRP1, impairs DRP1 higher order assembly and GTPase activity (Chang et al, 2010), but whether alternative molecular mechanisms drive mitochondrial hyperfusion and pathology caused by other pathological *DNM1L* variants, particularly affecting different domains, remains unknown.

Mitochondrial disease can arise from de novo heterozygous (Waterham et al, 2007; Chang et al, 2010; Chao et al, 2016; Fahrner et al, 2016; Sheffer et al, 2016; Vanstone et al, 2016; Zaha et al, 2016; Gerber et al, 2017; Whitley et al, 2018; Batzir et al, 2019; Vandeleur et al, 2019; Verrigni et al, 2019; Longo et al, 2020; Liu et al, 2021; Wei & Qian, 2021), biallelic compound heterozygous (Nasca et al, 2016; Yoon et al, 2016; Hogarth et al, 2018; Verrigni et al, 2019), and homozygous recessive (Hogarth et al, 2018) *DNM1L* variants (Table S1). The clinical course of individuals harbouring de novo *DNM1L* variants is both variable and unpredictable. Although there are no clear parallels between the clinical presentations and location of reported *DNM1L* variants, some patterns in genotype–phenotype correlations are starting to emerge. Over time, we anticipate that an increased mechanistic understanding of how *DNM1L* variants cause mitochondrial hyperfusion will enable us to understand whether specific variants may be amenable to therapeutic intervention.

Using massively parallel sequencing techniques, we identified five unrelated patients harbouring four previously unreported de novo heterozygous variants in *DNM1L*. Patients presented with a spectrum of neurological symptoms, as well as rarely reported cardiomyopathy, a clinical feature recapitulated in cardiac-specific *Dnm1l*⁻/⁻ knockout mice (Ikeda et al, 2015). Extensive in vivo and in vitro functional characterisation of patient *DNM1L* variants demonstrate that they impair mitochondrial network maintenance and peroxisomal morphology via divergent mechanisms, with variants in the DRP1 stalk domain correlating to greater disease severity and earlier age of death. We found that distinct *DNM1L* variants either increased or diminished GTPase activity, altered protein stability and impaired oligomerisation in the aetiology of *DNM1L*-related mitochondrial disease, subsequently leading to impaired mitochondrial and peroxisomal recruitment with organellar hyperfusion and functional deficiencies. In addition, we show that the p.Arg710Gly DRP1 GTPase effector domain (GED) variant can impair assembly driven GTP hydrolysis through disruption of the highly conserved hinge 1 region in a human dynamin related protein. Uniquely, this variant uncouples DRP1 oligomerisation from assembly-stimulated GTP hydrolysis, giving us a powerful tool to investigate how signals are transmitted from assembly state to the GTPase domain in dynamin-related proteins.

# Results

## Clinical data

We identified five individuals (patient 1 [P1], patient 2 [P2], patient 3 [P3], patient 4 [P4] and patient 5 [P5]) from five unrelated non-consanguineous families (Fig 1A) with developmental delay (four patients), a broad range of neurological manifestations including epilepsy (three patients), hypotonia (two patients), and/or cardiac problems (two patients). The detailed clinical findings of all five patients are described in the Supplemental Data 1 and Table 1.

## Molecular genetics investigations identify novel de novo heterozygous variants in *DNM1L*

To uncover candidate disease-causing variants in P1–P5, we used massively parallel sequencing techniques. Mitochondrial DNA

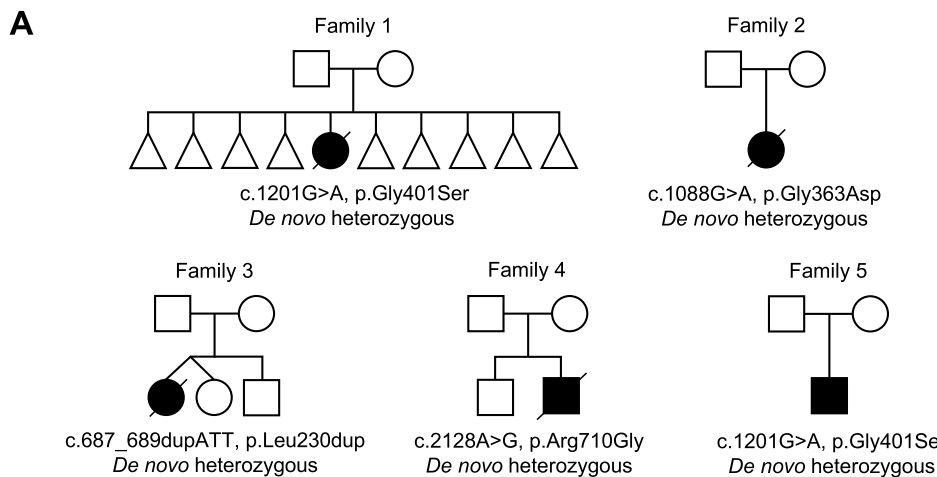

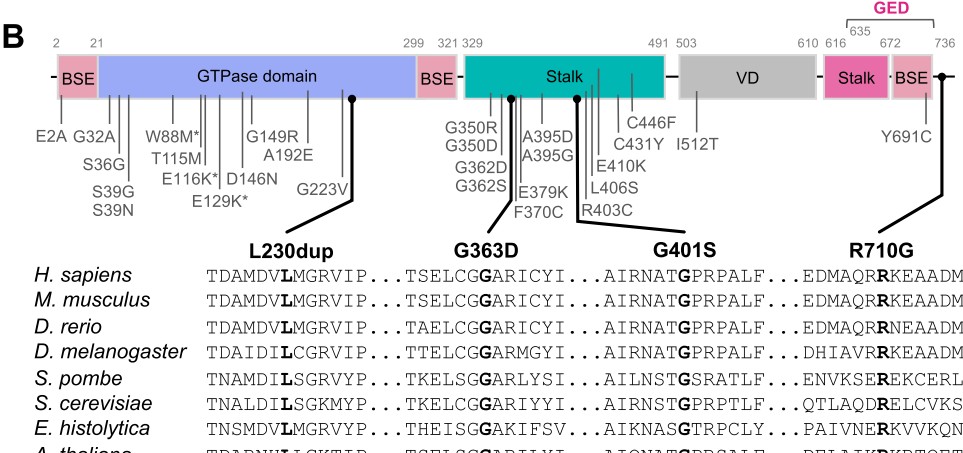

**Figure 1. Identification of five individuals harbouring de novo pathogenic variants in *DNM1L*.**
**(A)** Family pedigrees of *DNM1L* patients. Affected individuals are shown in black, squares represent males, circles represent females, triangles represent pregnancy not carried to term, and a diagonal line through the symbols indicates deceased subjects. **(B)** Schematic representation of known DRP1 variants and DRP1 protein domain organization: BSE (bundle signalling element), GTPase domain, stalk domain, variable domain (VD), and the GTPase effector domain (GED). Variants identified in this study are shown in black and previously reported variants are in grey. Partial amino acid sequence alignments of DRP1 showing evolutionary conservation across different species.

(mtDNA) genome sequencing of blood-derived DNA from P1 did not identify any likely pathogenic variants, whereas mtDNA copy number analysis using muscle-derived DNA found no evidence of mtDNA depletion. Trio array comparative genomic hybridization (aCGH) revealed a 15–20-kb chromosome 17p13.3 microdeletion of uncertain significance within an intronic region of YWHAE, but this was shown to be inherited from the father. Diagnostic whole exome sequencing (WES) analysis of the patient/parent trio identified a de novo heterozygous c.1201G>A, p.Gly401Ser *DNM1L* variant (NM_012062.5). The de novo heterozygous *DNM1L* c.1201G>A, p.Gly401Ser missense variant was classified as "likely pathogenic" using the Association of Clinical Genomic Science (ACGS) and The American College of Medical Genetics and Genomics (ACMG) guidelines (Richards et al, 2015) (https://www.acgs.uk.com/media/11631/uk-practice-guidelines-for-variant-classification-v4-01-2020.pdf) to apply the following criteria: PS2_moderate, PS3_moderate, PM2_moderate, PM4_supporting, and PP4_supporting.

Analysis of muscle DNA from P2 showed no evidence of mtDNA copy number abnormalities or mtDNA rearrangements, whereas sequencing of the entire mtDNA genome revealed no variants of pathological significance. On account of the apparent respiratory chain defect involving complex I, a targeted Ampliseq capture was used to facilitate analysis of the coding regions of the known nuclear-encoded complex I subunits and assembly factors (50 genes). Annotation and filtering of patient variants was performed as previously described (Alston et al, 2016) and identified a single, novel heterozygous variant c.152G>A, p.Arg51Gln in *NDUFS5* (NM_004552.3), which encodes a structural subunit of complex I. The c.152G>A, p.Arg51Gln variant was initially categorised as a "variant of uncertain significance" according to the ACGS/ACMG criteria PS2_moderate, PM2_moderate, PS3_supporting, PP3_supporting, and PP4_supporting. Patient cDNA studies showed no other variants in the fibroblast-derived *NDUFS5* cDNA transcript. Analysis of parental samples by Sanger sequencing supported a de novo occurrence. Concurrent unbiased trio WES analysis of P2 and her parents was performed which revealed an additional de novo heterozygous variant, c.1088G>A, p.Gly363Asp in *DNM1L*. This variant was classified as "likely pathogenic" using the ACGS/ACMG criteria PS2_moderate, PS3_supporting, PM2_moderate, PP3_supporting, and PP4_supporting. In light of the c.1088G>A p.Gly363Asp *DNM1L* variant identified in P2, the c.152G>A p.Arg51Gln *NDUFS5* variant was subsequently reinvestigated—4 heterozygote individuals are now recorded on gnomAD (two of which are adults) which is contra-indicative of a dominantly acting pathogenic variant meaning that

Table 1. Clinical, genetic, and pathological findings in individuals with *DNM1L* variants.

| ID | *DNM1L* variants | | | | Clinical features | Muscle biopsy and laboratory findings | |
|---|---|---|---|---|---|---|---|
| | cDNA (NM_012062.5) Protein (NP_036192.2) | Age-at-onset | Clinical course | Consanguinity; country of origin | Clinical features and relevant biochemical findings | Diagnostic muscle biopsy findings | Diagnostic biochemical findings |
| Patient 1[a] female | c.1201G>A, p.(Gly401Ser) de novo heterozygous | 8 mo | Died, 10 mo | No; UK | Seizures, developmental delay, microcephaly, sudden deterioration in feeding and breathing, brain MRI normal, ECG and echocardiogram abnormal, end-stage dilated cardiomyopathy with previous signs of hypertrophic cardiomyopathy, raised 3-MGA type IV, and plasma lactate 7.0 mmol/l (normal range 0.7–2.1 mmol/l) | Hyperfused and enlarged mitochondria, abnormal mitochondrial morphology with low cristae density on TEM | Low complex IV ratio of 0.010 (0.014–0.034) in muscle |
| Patient 2[a,b] female | c.1088G>A, p.(Gly363Asp) de novo heterozygous | Birth | Died, 13 mo | No; UK | Seizures, growth failure, developmental delay, failure to thrive, microcephaly, micrognathia, infantile spasms, hypotonia, brain MRI abnormal, electroencephalogram abnormal—hypsarrhythmia, echocardiogram showed mild left ventricular hypertrophy, CSF lactate 4.6–7.0 mmol/l (normal range 0.7–2.1 mmol/l) | n.d. | Complex I–immunodeficient muscle fibres (IHC) and low complex I and II respiratory chain complex activities in muscle; low complex I activities in fibroblasts |
| Patient 3[c] female | c.687_689dupATT, p.(Leu230dup) de novo heterozygous | 6 yr | Died, 20 yr | No; UK, Caucasian | Learning difficulties, epilepsy, ataxia, dystonia, myoclonus and peripheral neuropathy, blood and CSF lactate normal, glucose concentrations normal, urine organic acid and plasma amino acid analysis normal | Muscle electron microscopy and skin histology were not conclusive, but mainly normal | Complexes I–IV normal in the 1st muscle biopsy. 2nd muscle biopsy 3 yr later showed decreased complex I and IV activity |
| Patient 4[a] male | c.2128A>G, p.(Arg710Gly) de novo heterozygous | 3 yr | Died, 17 yr | No; UK | Chronic inflammatory demyelinating polyneuropathy, extra-pyramidal movement disorder, epilepsy, optic atrophy, fatigue, and episodic regression of developmental skills precipitated by infection | n.d. | Mitochondrial respiratory chain activities (complexes I–IV) in muscle normal |
| Patient 5[d] male | c.1201G>A, p.(Gly401Ser) de novo heterozygous | 33 mo | Alive, 3 yr | No; UK Caucasian | Early onset epileptic encephalopathy, global developmental delay, hypotonia, nystagmus, dyskinesia, lactate and pyruvate concentrations in the CSF normal, plasma amino acids, urinary amino acids, organic acids and urine sialic acid normal | n.d. | n.d. |

[a]Investigated by trio whole exome sequencing.
[b]Investigated by mitochondrial gene panel.
[c]Investigated by 100,000 genome project.
[d]Investigated by WES.

the PM2 criterion is no longer applicable. Moreover, in light of an alternative diagnosis (*DNM1L*-related disease), the guidelines support application of the BP5 criterion which reclassifies the c.152G>A p.Arg51Gln *NDUFS5* variant as "likely benign."

Initial investigations for P3 including mtDNA genome analysis and mtDNA copy number analysis were normal. P3 was subsequently enrolled onto the Genomics England 100,000 genome sequencing project, with targeted data analysis focusing on the gene panels for hereditary ataxia (v1.51) and paediatric motor neuronopathies (v1.0). Comparative genomic hybridization assay revealed a chromosome 19p13.3 microduplication that was not present in either parent, but its significance was uncertain. This

analysis identified a single heterozygous c.687_689dup, p.Leu230dup *DNM1L* variant and analysis of parental samples supported a de novo occurrence. The 687_689dup, p.Leu230dup variant was classified as "likely pathogenic" using the ACGS/ACMG criteria PS2_moderate, PS3_moderate, PM2_moderate, PM4_supporting, and PP4_supporting.

Initial diagnostic investigations for P4 excluded the presence of common pathogenic *POLG* variants or a pathogenic mtDNA variant. Subsequently, trio WES analysis of P4 and his parents identified a single heterozygous c.2128A>G, p.Arg710Gly *DNM1L* variant that had arisen de novo in the proband. The c.2128A>G, p.Arg710Gly variant was classified as "likely pathogenic" using the ACGS/ACMG criteria PS2_moderate, PS3_moderate, PM2_moderate, PP3_supporting, and PP4_supporting.

Finally, DNA from P5 was subject to singleton WES analysis which revealed the same single heterozygous c.1201G>A, p.Gly401Ser *DNM1L* variant that was present in P1.

All *DNM1L* variants have not been previously reported pathogenic and were absent from gnomAD database (https://gnomad.broadinstitute.org/). The *DNM1L* variants were confirmed by Sanger sequencing, and analysis of parental samples was undertaken either as part of the trio WES pipeline, or by targeted Sanger sequencing which supported the de novo occurrence of a *DNM1L* variant in each clinically affected child.

### In silico structural modelling of DRP1 variants

Three of the five patients (P1, P2, and P5) exhibited single-nucleotide variations, c.1201G>A, p.Gly401Ser (G401S), or c.1088G>A, p.Gly363Asp (G363D), in the DRP1 stalk domain (Fig 1B) which has been shown to play a key role in dimerization and self-assembly essential for fission (Fröhlich et al, 2013; Francy et al, 2017; Kalia et al, 2018). Analysis of the cryoEM structure of DRP1 in co-complex with one of its recruiting proteins, MID49 (PDB:5WP9), suggests that both residues are located at the dimer interface (Fig 2A). Indeed, a quadruple mutant G401-404 AAAA has been shown to promote disruption of tetramers (or any higher order oligomers) and the formation of stable dimers under certain conditions for DRP1 and other dynamin related proteins (Gao et al, 2010, 2011; Faelber et al, 2011; Ford et al, 2011; Fröhlich et al, 2013). The shared variant in P1 and P5 involves residue G401 which serves as a C-terminal capping residue for $\alpha$-helix 1 in the stalk domain. Glycine is the most common C-terminal capping residue as it can adopt a wide range of $\varphi$ $\psi$ angles because of its small, single hydrogen-containing R-group, allowing for termination of a helix (Richardson & Richardson, 1988; Aurora et al, 1994; Bang et al, 2006; Beck et al, 2008). In the 5WP9 structure, G401 adopts a $\varphi$ angle of 78.3° and $\psi$ angle of −160° (Fröhlich et al, 2013), a generally unfavourable conformation for residues other than glycine, which likely allows it to form a sharp helix-turn-helix, a prevalent structural motif in DRP1. Conversely, serine has a limited number of preferred $\varphi$ $\psi$ angles (Beck et al, 2008) and a G401S substitution would likely result in an energetically unfavourable eclipsed conformation of the R group and adjacent amino or carbonyl groups. This would almost certainly introduce significant steric clashes, slightly destabilize the helix, and may impact self-assembly.

Regarding P2, G363 is an N-terminal $\alpha$-helix capping residue and is in close proximity (4.2 Å) to the G401 residue of a neighbouring monomer (Fig 2B). Like the G401S substitution described above, G363 has relatively uncommon $\varphi$ $\psi$ angles of −107.6° and −82.6°, respectively. The substitution of G363 to a larger charged aspartic acid, which does not typically populate those $\varphi$ $\psi$ angles (Beck et al, 2008), would likely induce significant steric clashes with several nearby residues, including G401 and P402 (inter-molecular clashes) and E349 (intra-molecular clash). This could in turn disturb local secondary structure because of $\alpha$-helix destabilisation, as well as DRP1 dimerisation. However, given the residue is adjacent to a flexible loop, one could predict that this region may be able to accommodate minor structural changes with no effects on dimer stability.

In P3, there is an insertion of an extra leucine at position 231 (L230dup) within the GTPase domain (Fig 1B), in a short $\alpha$-helix that is flanked by two disordered loops, the canonical G4 (N-terminal of Leu230) and G5 (C-terminal of L230) motifs. The G4 and G5 motifs (Fig 2C) are critical for nucleotide binding (Wenger et al, 2013), and it is possible that the L230 duplication transmits a conformational change to these proximal loops and critical nucleotide binding residues such as K216, D218, and N246, impacting their GTP binding ability. In addition, dimerization via the GTPase domain is essential for GTP hydrolysis, and L230/L231 is spatially located near the $\alpha$-helix containing the critical dimerization residue D190 (Kishida & Sugio, 2013; Wenger et al, 2013). Introduction of the extra leucine at position 231 has the potential to introduce conformational changes in nearby regions, such as the adjacent G4 and G5 motifs or to the D190 containing helix, which may ultimately impair GTPase domain dimerization. Furthermore, the areas surrounding the L230/L231 residues of wild-type DRP1 engage in an extensive interface with MID49 (Fig 2D) (Kalia et al, 2018). This interface is also mediated in part by the N-terminal loop of this region, specifically residue D221 of the G4 loop, which may be impaired by the L230 duplication. Altogether, these predictions suggest multiple mechanisms by which the L230dup event may lead to impaired DRP1 activity.

In P4, the residue R710 is located within the bundle signalling element domain (Fig 1B), a highly conserved position among the dynamin superfamily (Muhlberg et al, 1997; Sever et al, 1999; Zhu et al, 2004; Gao et al, 2010, 2011; Faelber et al, 2011; Ford et al, 2011; Fröhlich et al, 2013). R710 forms a salt bridge with E702 in the C-terminal loop L2$^{BS}$ which is part of a highly conserved hinge motif between the GTPase and stalk domains (Fröhlich et al, 2013). Substitution of this charged arginine to a small non-polar glycine would induce a loss of this salt bridge, likely leading to decreased protein stability and altered conformation of the hinge (Fig 2E). In dynamin and the human myxovirus resistance protein 1 (MxA), both of which belong to the dynamin superfamily of large GTPases, the hinge region is thought to facilitate conformational changes that lead to assembly stimulated GTP hydrolysis (Sever et al, 1999; Gao et al, 2011; Fröhlich et al, 2013). Crystallographic structural data of DRP1 revealed monomers with two different conformations, differing in their positioning of the GTPase domain and bundle signalling element in relation to the stalk, suggesting that similar large-scale conformational changes around this hinge region are possible and may relay assembly information to the GTPase domain in a similar manner (Fröhlich et al, 2013). Therefore, the disruption in stability would likely have a negative impact on DRP1 assembly-stimulated hydrolysis.

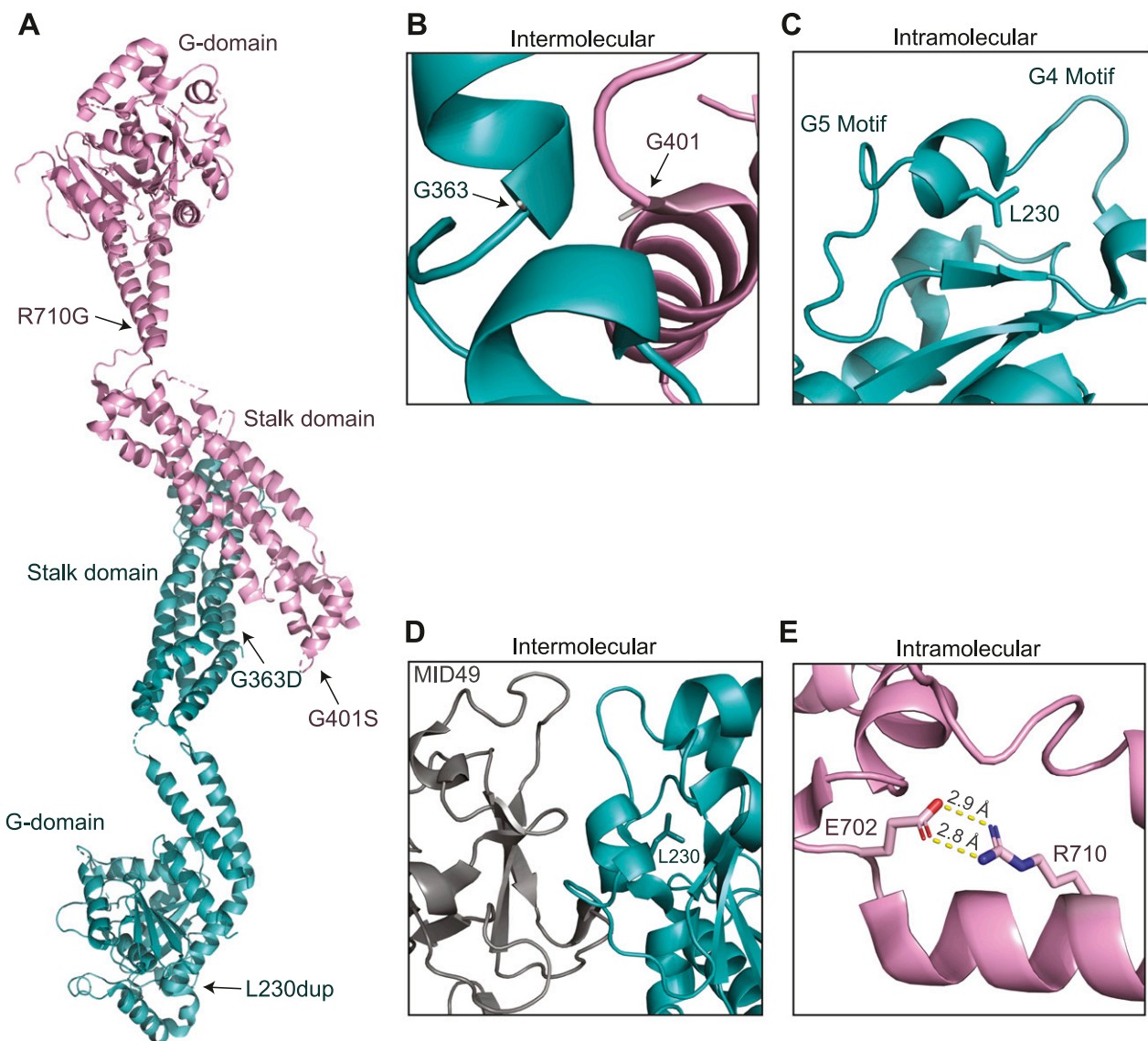

**Figure 2.   In silico structural studies of DRP1 variants.**
**(A)** Locations of pathogenic variants marked on the crystal structure of nucleotide-free DRP1 (PDB: 4BEJ). **(B, C, D)** Residue–residue interactions and spatial relationships of residues to neighbouring motifs or DRP1 monomers of the wild-type version of residues from (A) (CryoEM structure of DRP1 assembled and in complex with MID49, PDB: 5WP9). **(B)** Both G363 and G401 are α-helix capping residues found in close-proximity to each other between neighbouring DRP1 monomers. Substitution of either glycine to a charged aspartate (G363D) or polar serine (G401S) would induce unfavourable steric clashes with neighbouring residues and disrupt helix stability. **(C)** L230 is located within a small α helix between the G4 and G5 loop motifs, critical for nucleotide binding. Addition of another leucine to this helix may disrupt these motifs, impairing GTP binding. **(D)** The helix containing L230 is adjacent to the MID49 binding surface and the L230 duplication in this location may have negative effects on MID49 binding and recruitment of DRP1 to the mitochondria. **(E)** The residue R710G, located within the bundle signalling element domain, forms a salt bridge with E702. The R710G substitution would induce a loss of this salt bridge.

## Mitochondrial and peroxisomal network analysis of *DNM1L* patient fibroblasts

Impaired mitochondrial fission due to defective DRP1 results in altered mitochondrial networks that are characterised by elongated and highly interconnected filamentous mitochondria. To assess the impact of the *DNM1L* variants identified in P1 (p.Gly401Ser), P2 (p.Gly363Asp), P3 (p.Leu230dup), and P4 (p.Arg710Gly) on mitochondrial morphology, live mitochondrial networks in available patient-derived fibroblasts were visualised using high-resolution confocal imaging after incubation with tetramethylrhodamine (TMRM), a cell-permeant dye that is actively sequestered into mitochondria on the basis of the membrane potential.

Analysis of mitochondrial networks using the ImageJ tool Mitochondrial Network Analysis (MiNA) revealed marked hyperfusion of mitochondria in P1, P2, and P4 compared to age-matched controls (Fig 3A and B). In addition, the mitochondrial network length was analysed using immunofluorescence labelling of fixed patient and age-matched control fibroblasts using TOM20 antibodies. The Columbus (PerkinElmer) software system was used to

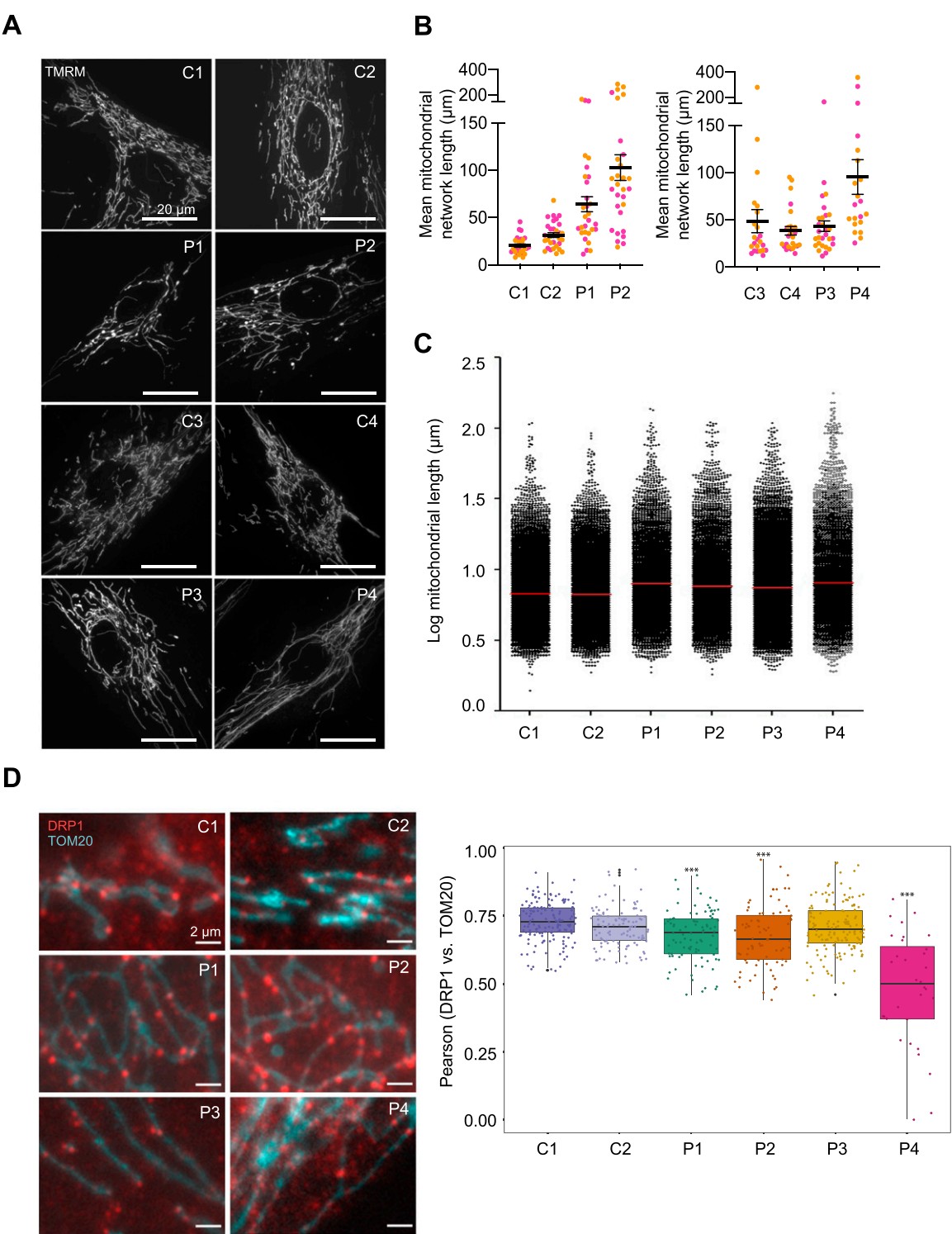

**Figure 3. The impact of *DNM1L* variants on mitochondrial network length and DRP1 mitochondrial co-localisation.**
**(A)** Representative images of TMRM-stained mitochondrial network in paediatric (C1 and C2) and adult (C3 and C4) controls and *DNM1L* patient (P1–P4) fibroblasts.
**(B)** Quantification of mean mitochondrial network length via MiNa using ImageJ > 20 fields from two independent experiments, calculated by multiplying mean branch length and mean number of branches per network. Non-parametric one-way ANOVA and Dunn's multiple comparisons using GraphPad Prism were used to calculate statistically significant differences between groups. **(C)** Mitochondrial network length using immunofluorescence analysis of fixed paediatric control (C1), adult control (C2), and *DNM1L* patient (P1–P4) fibroblasts labelled with TOM20 antibodies. The Columbus (PerkinElmer) software was used to quantify the hyperfusion of patient mitochondrial networks relative to controls and a minimum of 5,500 mitochondria were analysed for each case. The immunofluorescence labelling was performed three times. **(D)** Analysis of DRP1 co-localisation with the outer mitochondrial membrane protein TOM20 by immunofluorescence labelling of age-matched controls (C1: paediatric, C2: adult) and *DNM1L* patient (P1–P4) fibroblasts with anti-DRP1 (red puncta) and anti-TOM20 (in blue). DRP1 co-localisation with mitochondria was analysed in

quantify the hyperfusion of patient mitochondrial networks relative to controls. A minimum of 5,500 mitochondria were analysed for each case. Largely consistent with live cell imaging, significant hyperfusion of mitochondrial networks were observed in all four studied patient fibroblasts using this approach (Fig 3C). Whereas live cell imaging did not reveal extensive mitochondrial hyperfusion in P3 fibroblasts, TOM20 immunostaining revealed elongated mitochondria in P3 (p.Leu230dup) cells. Notably, these cells were the least affected compared with those from other patients (Fig 3C).

To determine whether mitochondrial network alterations were due to decreased DRP1 recruitment, we performed a co-localisation analysis using the Pearson's co-localisation coefficient between DRP1 and TOM20 which showed decreased DRP1 at the mitochondria in P1 (p.Gly401Ser), P2 (p.Gly363Asp), and P4 (p.Arg710Gly) fibroblasts. Of these, P4 (p.Arg710Gly) had the most severe recruitment defect with the lowest Pearson's R value and DRP1 appearing primarily cytosolic without punctate structures, which were still seen in other variants albeit to a lesser extent than the control fibroblasts (Fig 3D).

Although the degree of mitochondrial hyperfusion differed between patient fibroblasts, with P3 (p.Leu230dup) not displaying significant elongation by MiNA, this phenotype was consistent with previously reported de novo heterozygous *DNM1L* variants (c.95G>C, p.Gly32Ala; c.436G>A, p.Asp146Asn; c.1184C>A, p.Ala395Asp; c.1207C>T, p.Arg403Cys; c.1292G>A, p.Cys431Tyr) and a GTPase-deficient recombinant mutant (p.Lys38Ala) (Zhu et al, 2004; Waterham et al, 2007; Chang et al, 2010; Whitley et al, 2018; Longo et al, 2020).

Given DRP1 has been implicated in both mitochondrial and peroxisomal fission (Waterham et al, 2007), we examined the effect of these variants on peroxisomal networks. Immunofluorescence labelling of control and *DNM1L* patient fibroblasts with antibodies against the peroxisomal membrane protein marker PMP70 was used to determine the peroxisomal morphology. The analysis using the Columbus software revealed that peroxisomes in P1 (p.Gly401Ser), P2 (p.Gly363Asp), P3 (p.Leu230dup) and P4 (p.Arg710Gly) appeared more fused with fewer overall numbers of peroxisomes and decreased size distribution, indicative of impaired fission (Fig 4A).

Co-localisation analysis between DRP1 and PMP70 showed decreased DRP1 at the peroxisomes in P1 (p.Gly401Ser) and P2 (p.Gly363Asp), but not P3 (p.Leu230dup) and P4 (p.Arg710Gly), suggesting that the elongated peroxisomes in P3 and P4 are not simply due to decreased DRP1 recruitment (Fig 4B). Previous reports argue that not all *DNM1L* variants impair peroxisomal morphology, with several other variants in the GTPase domain having no impact on peroxisomal morphology despite affecting mitochondrial network morphology. Specifically, the p.Glu2Ala, p.Ala192Glu (Gerber et al, 2017), and p.Gly32Ala (Whitley et al, 2018) variants had normal peroxisomes in the setting of abnormal mitochondrial networks. Conversely, patient fibroblasts from a biallelic heterozygous patient carrying p.Ser36Gly; p.Glu116Lysfs*6 variants had both abnormal peroxisomal and mitochondrial fission (Nasca

et al, 2016). Similar impairments were also observed in the p.Asp146Asn (Longo et al, 2020) and p.Gly223Val variants (Verrigni et al, 2019) (Table S1).

## Mitochondrial DNA nucleoid analysis of de novo *DNM1L* variants

Defective mitochondrial fission has also been associated with the formation of enlarged bulb-like structures ("mito-bulbs") caused by nucleoid clustering (Ban-Ishihara et al, 2013). Previously, *DNM1L* siRNA knockdown in HeLa cells as well as *Dnm1l*$^{-/-}$ knockout mice studies have demonstrated severe mtDNA nucleoid aggregation within the hyperfused mitochondrial networks, leading to respiratory deficiency and heart dysfunction in the fission-deficient mice (Ban-Ishihara et al, 2013; Ishihara et al, 2015). Imaging of fibroblasts incubated with TMRM revealed the presence of enlarged mitochondria in all patients (Fig S1A), with P1 (p.Gly401Ser) and P2 (p.Gly363Asp) most widely affected. Subsequent co-staining of P1 and P2 fibroblasts with TMRM and PicoGreen (a fluorochrome which reveals nucleoids by illuminating mtDNA) demonstrated the co-localisation of these enlarged "mito-bulbs" with large nucleoids (Fig S1B). Detailed analysis of mtDNA nucleoids stained with PicoGreen using Columbus software (PerkinElmer) revealed marked differences in the proportion of enlarged nucleoids (area > 1.5 $\mu m^2$) in P1 (p.Gly401Ser) and P2 (p.Gly363Asp) compared with control (Fig S1C). There was no difference in nucleoid size ratio between P3 (p.Leu230dup) and control (Fig S1C). Although, upon visual examination P4 (p.Arg710Gly) nucleoids appeared enlarged compare with controls, we were not able to accurately quantify the individual puncta because of increased levels of lipofuscin present in these cells (Fig S1D).

Altogether, assessment of patient fibroblasts demonstrated that the de novo variants identified in P1 (p.Gly401Ser) and P2 (p.Gly363Asp) cause mitochondrial network hyperfusion, leading to mitochondrial enlargement and nucleoid clustering which is indicative of impaired nucleoid distribution and segregation.

## The effect of *DNM1L* variants on DRP1 protein expression

To evaluate the molecular consequences of the c.1201G>A, p.Gly401Ser; c.1088G>A, p.Gly363Asp; c.687_689dupATT, p.Leu230dup, and c.2128A>G, p.Arg710Gly DRP1 variants, primary patient fibroblasts (P1–P4) and age-matched controls (C1–C4) were analysed by SDS–PAGE and immunoblotting (Fig S2). Normal levels of DRP1 protein in the monomeric form were found in P1 (p.Gly401Ser), P2 (p.Gly363Asp), and P3 (p.Leu230dup), whereas P4 (p.Arg710Gly) showed decreased levels of DRP1 when compared with controls (Fig S2). These data suggest that the mutated p.Gly401Ser, p.Gly363Asp, and p.Leu230dup DRP1 protein is expressed in P1, P2, and P3, respectively, and may act in a dominant-negative fashion, overriding the effect of the wild-type allele. DRP1 recruitment to the mitochondrial membrane is dependent on adaptor proteins such as MID49 and MID51. However, their role in disease remains largely

at least 32 cells per subject in two independent experimental sets. Pearson's correlations between DRP1 puncta and TOM20 in each cell line are shown as box plots. One-way ANOVA with post hoc Tukey's honest significant difference test was used to determine statistically significant differences (***$P \leq 0.001$). Representative merged immunofluorescence images of fibroblasts stained with anti-TOM20 and anti-DRP1 antibodies are shown on the left.

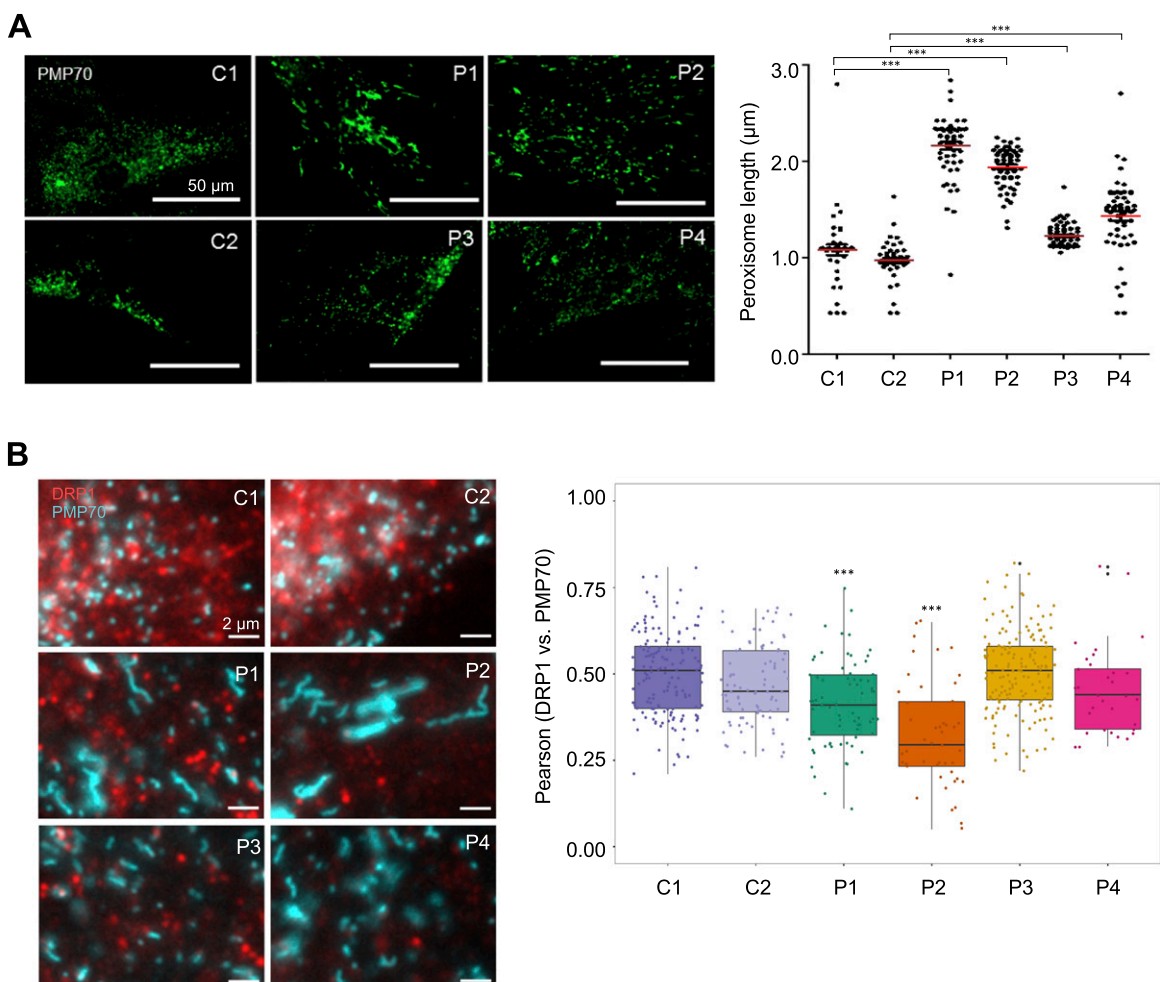

**Figure 4. The effect of *DNM1L* variants on peroxisomal morphology and co-localisation of DRP1 with peroxisomes.**
**(A)** Analysis of peroxisome length by immunofluorescence using a peroxisomal membrane marker (PMP70) in fixed age-matched controls (C1: paediatric, C2: adult) and *DNM1L* patient (P1–P4) fibroblasts. The Columbus (PerkinElmer) software was used to quantify the peroxisome length between patients and controls. The immunofluorescence labelling was performed three times and a minimum of 300 peroxisomes were analysed in each case. Statistically significant differences between groups were determined by a non-parametric one-way ANOVA (***$P \leq 0.001$). Representative images of fixed cells stained for peroxisomes (PMP70) in control (C1 and C2) and *DNM1L* patient (P1–P4) fibroblasts are shown on the left. **(B)** Immunofluorescence analysis of DRP1 puncta (red) co-localising with peroxisomes (PMP70 in blue) in age-matched control (C1: paediatric, C2: adult) and *DNM1L* patient (P1–P4) fibroblasts. The analysis was performed on at least 32 cells from two independent experimental sets and mean values showing Pearson's correlation between the proportion of DRP1 puncta and peroxisomal marker PMP70 are shown. Statistically significant differences were calculated via a one-way ANOVA with post-hoc Tukey's honest significant difference test (***$P \leq 0.001$). Representative merged immunofluorescence images of PMP70 and DRP1 stained cells are shown on the left.

unclear. It has recently been described that MID51 regulates the assembly and fission activity of DRP1 (Ma et al, 2019). Western blot analysis of *DNM1L* patient fibroblasts revealed that levels of MID51 are similar in both patient and control fibroblasts (Fig S2), suggesting that the *DNM1L* variants do not affect the stability of the MID51 adaptor protein.

### Diagnostic histological and biochemical investigations

Diagnostic respiratory chain enzyme analysis of cytochrome *c* oxidase (COX) and succinate dehydrogenase in P1 (p.Gly401Ser) muscle revealed decreased complex IV activity (Table 1). A quadruple immunofluorescent (IHC) assay, which quantifies protein levels of COX1, NDUFB8, porin, and laminin in individual myofibres

(Rocha et al, 2015), detected complex I-immunodeficient muscle fibres in P2 (p.Gly363Asp) (Fig S3). In addition, diagnostic spectrophotometric biochemical measurements of mitochondrial respiratory chain complex activities in the available muscle from P2 (p.Gly363Asp), P3 (p.Leu230dup), and P4 (p.Arg710Gly) were determined (Fig S4A). P2 (p.Gly363Asp) muscle showed decreased activities of complex I and complex II, whereas the activities of complexes III and IV were normal (Fig S4A). Two separate muscle biopsies have been taken in P3 (p.Leu230dup) at the age of 13 and 16 yr, respectively. The spectrophotometric respiratory chain complex activities were normal in the first muscle biopsy; however, the latter one showed a complex I and complex IV deficiency, suggesting a progressive defect (Fig S4A). Mitochondrial respiratory chain activities in P4 (p.Arg710Gly) skeletal muscle were normal,

except for increased complex III activity, which may be attributed to a compensatory response mechanism (Fig S4A).

## Variants in *DNM1L* impair levels of OXPHOS proteins

Next, we determined whether the mitochondrial network anomalies present in *DNM1L* patient fibroblasts were associated with OXPHOS dysfunction. Western blotting and quantification of bands obtained by densitometry analysis of P1 (p.Gly401Ser) fibroblasts revealed that the steady-state levels of OXPHOS proteins were relatively normal, except for mild decreases in the complex I subunit, NDUFB8 and the complex IV subunit COX2 (Fig S5), which was consistent with the observed decreased complex IV activity in muscle tissue (Table 1). P2 (p.Gly363Asp) mutant fibroblasts presented with a decrease in NDUFB8, UQCRC2, and COX2 protein levels (Fig S5). In addition, the marked decrease in NDUFB8 protein levels detected by Western blotting correlate with the impaired complex I activity in patient-derived muscle and fibroblasts (Fig S4A and B). NGS analysis of P2 also identified a de novo heterozygous c.152G>A, p.Arg51Gln variant in the *NDUFS5* gene encoding a core accessory subunit of complex I. The c.152G>A, p.Arg51Gln *NDUFS5* variant could partially contribute to the decreased levels of NDUFB8 protein and impaired complex I activity; however, in silico pathogenicity assessment classified the variant as likely benign and not pathogenic. A multiple OXPHOS defect was present in P3 (p.Leu230dup) fibroblasts, showing decreased steady-state levels of NDUFB8, UQCRC2, and COX2 (Fig S5), where only impaired complex I and complex IV activity, correlated with the respiratory chain measurements in muscle (second biopsy) (Fig S4A). Furthermore, a decrease in complex I (NDUFB8) and complex IV (COX2) subunits was detected in P4 (p.Arg710Gly) fibroblasts when compared with controls (Fig S5). Similar to the increased complex III activity in P4 muscle tissue (Fig S4A), densitometry analysis of the complex III subunit in P4 fibroblasts showed mild increase in the steady-state levels of UQCRC2 (Fig S5).

Interestingly, there are some differences between the OXPHOS abnormalities amongst the patient muscle samples and fibroblasts. Most notably, P4 (p.Arg710Gly) whom had increased complex III activity in skeletal muscle, but decreased complex I and IV proteins in fibroblasts. We hypothesize that these differences likely stem from tissue-specific effects on respiration. Together these data suggest that different *DNM1L* variants have distinct impact on OXPHOS function in fibroblasts, with minimal correlations to disease onset or severity, suggesting that the OXPHOS defects present in cells are a secondary consequence of the disrupted mitochondrial network balance as opposed to a driver of disease.

## Patient DRP1 variants have altered GTPase activity

DRP1 performs its mechanoenzyme function of mitochondrial membrane constriction through the hydrolysis of GTP following its assembly on the mitochondrial outer membrane. To determine whether *DNM1L* variants altered GTPase activity in vitro, we first expressed human DRP1 in recombinant form recapitulating the disease-causing variants identified in P1 and P5 c.1201G>A, p.Gly401Ser (G401S), P2 c.1088G>A, p.Gly363Asp (G363D), P3 c.687_689dupATT, p.Leu230dup (L230dup), and P4 c.2128A>G, p.Arg710Gly (R710G). Bacterial expression of all variants were-

similar to wildtype (WT), except for L230dup which did not produce any full-length protein under multiple conditions and was unable to be purified for further studies. Wild-type human DRP1 and the remaining variants were purified to homogeneity and found to be well folded by circular dichroism (Fig S6), but differences in the mean residue ellipticity suggested differences in structure that might affect GTP hydrolysis. To test this, GTP hydrolysis was measured in solution with increasing amounts of GTP substrate to determine the apparent Michaelis constant ($K_{0.5}$), the turnover number ($k_{cat}$), and catalytic efficiency ($k_{cat}/K_{0.5}$) (Fig 5A–D). The activity of WT enzyme was similar to previous measurements (Chang et al, 2010; Fröhlich et al, 2013; Koirala et al, 2013; Bustillo-Zabalbeitia et al, 2014; Cahill et al, 2015; Francy et al, 2017) with a $K_{0.5}^{GTP}$ of 201 ± 51 $\mu$M, a $k_{cat}$ of 0.24 min$^{-1}$, and $k_{cat}/K_{0.5}$ of 1.2 × 10$^{-3}$ $\mu$M$^{-1}$·min$^{-1}$ (Fig 5B–D and Table 2). These substrate kinetic experiments with DRP1 variants G363D, G401S, and R710G demonstrated modestly altered GTPase activity with R710G decreasing, and G363D and G401S increasing, the turnover number (Fig 5A). Curiously, each variant decreased the $K_{0.5}$ for GTP suggesting they modestly increased the catalytic efficiency for G363D and G401S, but not R710G. Overall, these data suggest that impaired hydrolysis is not a major factor in pathogenesis of patients harbouring these variants. Although not tested, we predict the duplication of L230 would be deleterious to GTPase activity given the potential for direct disruption to the adjacent G4 and G5 loop motifs involved in nucleotide binding, or potential shifting of the interaction domains of DRP1 (i.e., GTPase domain, GED, and/or stalk).

## Patient DRP1 variants have impaired self-assembly

DRP1 assembles in the cytoplasm and around the circumference of the mitochondrion to effect membrane scission. To evaluate the pathological variants ability to self-assemble, we used size-exclusion chromatography with multi-angle laser light scattering (SEC-MALS, Fig 6A). This method is advantageous over traditional size-exclusion chromatography as it allows for the direct determination of molecular weight instead of relying on comparisons to molecular weight standards with different molecular conformations that can influence their elution time (Some et al, 2019). Wild-type DRP1 was found to primarily elute in two peaks corresponding to dimeric (elution time ~9 min) and tetrameric (elution time ~8 min) populations (Fig 6A), consistent with previous findings (Chang et al, 2010; Fröhlich et al, 2013; Macdonald et al, 2016; Francy et al, 2017). The range of molecular weight species observed on the chromatogram were interpreted to be due to a dynamic exchange between the oligomeric states during elution.

Each of the patient DRP1 variants was found to impair higher order assembly to differing degrees, as determined by a decrease in the amount of higher molecular weight species eluting at earlier time points. Both G363D and G401S appear to be primarily dimeric, confirming previous studies on G363D (Tanaka et al, 2006; Chang et al, 2010; Clinton et al, 2016). Furthermore, each of the variants altered the elution profile in that there was only one primary peak versus the more complex elution profile of wild-type DRP1, suggesting that these substitutions alter the exchange rate between oligomeric species. Alterations to the elution profile and thus rates of exchange between species, have been observed before with

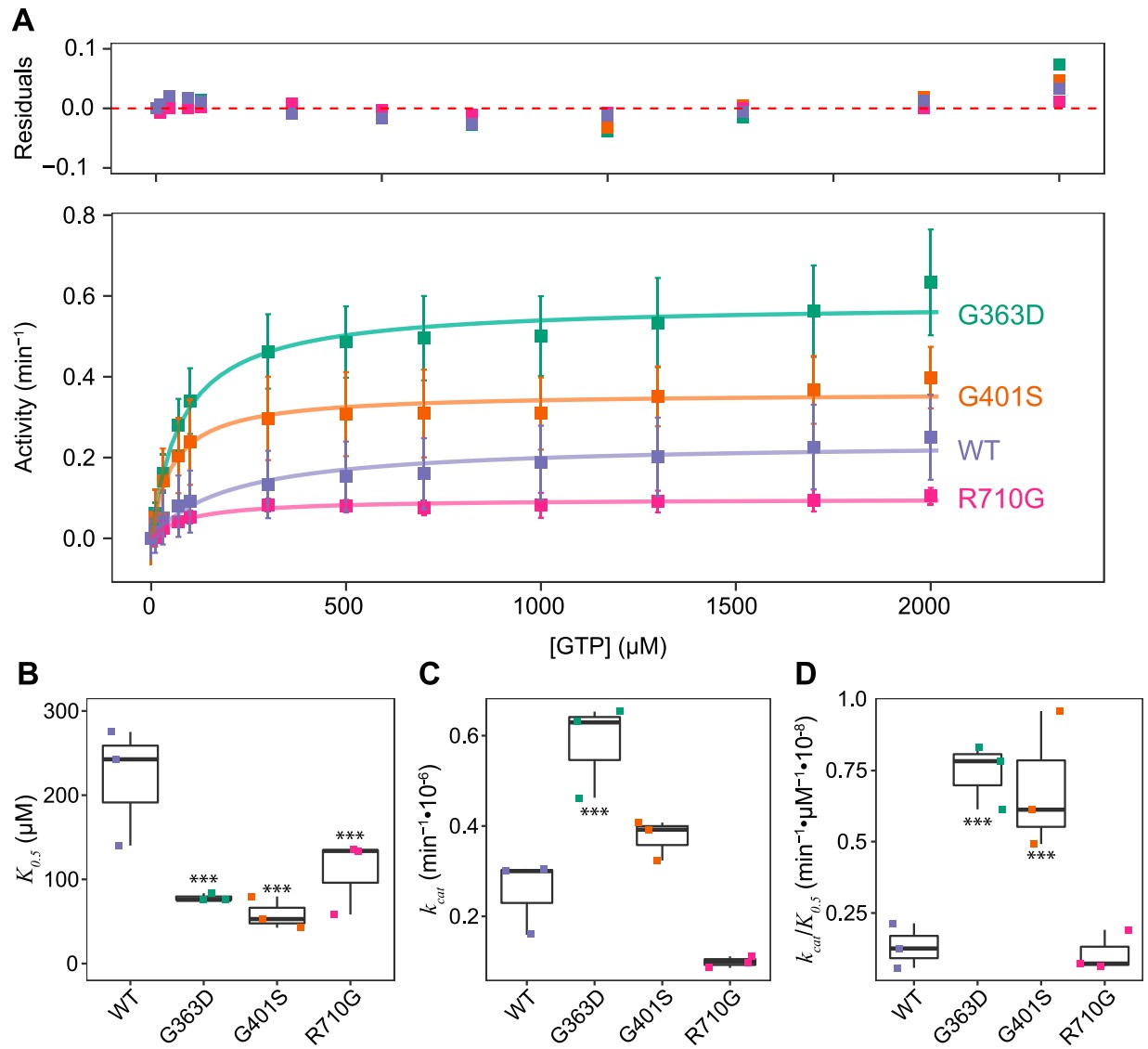

**Figure 5. Clinically identified *DNM1L* variants alter GTPase activity.**
**(A)** Substrate kinetics of recombinant wild-type DRP1 (WT) (1 $\mu$M) and genetic variants. DRP1 GTPase activity was measured using an enzyme coupled assay monitoring NADH depletion, which is subsequently converted to activity (min$^{-1}$). Data from three independent experiments were globally fit to a Michaelis–Menten model. Residuals of the fit are shown. **(B, C, D)** Distribution of $K_{0.5}$, (C) $k_{cat}$, and (D) $k_{cat}/K_{0.5}$ parameters from GTPase activity measurements. Reported values were obtained by globally fitting DRP1 GTPase activity measurements (n = 3) to a Michaelis–Menten model. The resulting values are reported in Table 2. $K_{0.5}$ differences between WT and each variant significant to ***$P < 0.05$. $k_{cat}$ differences between WT and G363D, G363D and R710G, G363D and G401S, and R710G and G401S significant to ***$P < 0.05$. $k_{cat}/K_{0.5}$ differences between WT and both G363D and G401S, as well as between R710G and both G363D and G401S significant to ***$P < 0.05$.

**Table 2. Reported kinetic values among DRP1 variants. Kinetic parameters ($K_{0.5}$, $V_{max}$, $k_{cat}$, and $k_{cat}/K_{0.5}$) were computed for DRP1 WT and each clinical variant.**

|       | $K_{0.5} \pm$ SD ($\mu$M) | $V_{max} \pm$ SD ($\mu$M/min) | $k_{cat}$ (min$^{-1}$) | $k_{cat}$ (min$^{-1}$)/$K_{0.5}$ ($\mu$M) |
|-------|---------------------------|-------------------------------|------------------------|-------------------------------------------|
| WT    | 201 ± 51                  | 0.24 ± 0.01                   | 0.24 × 10$^{-6}$       | 1.2 × 10$^{-9}$                           |
| G363D | 79 ± 11                   | 0.58 ± 0.02                   | 0.58 × 10$^{-6}$       | 7.3 × 10$^{-9}$                           |
| G401S | 55 ± 9                    | 0.36 ± 0.011                  | 0.36 × 10$^{-6}$       | 6.5 × 10$^{-9}$                           |
| R710G | 96 ± 18                   | 0.10 ± 0.004                  | 0.10 × 10$^{-6}$       | 1.0 × 10$^{-9}$                           |

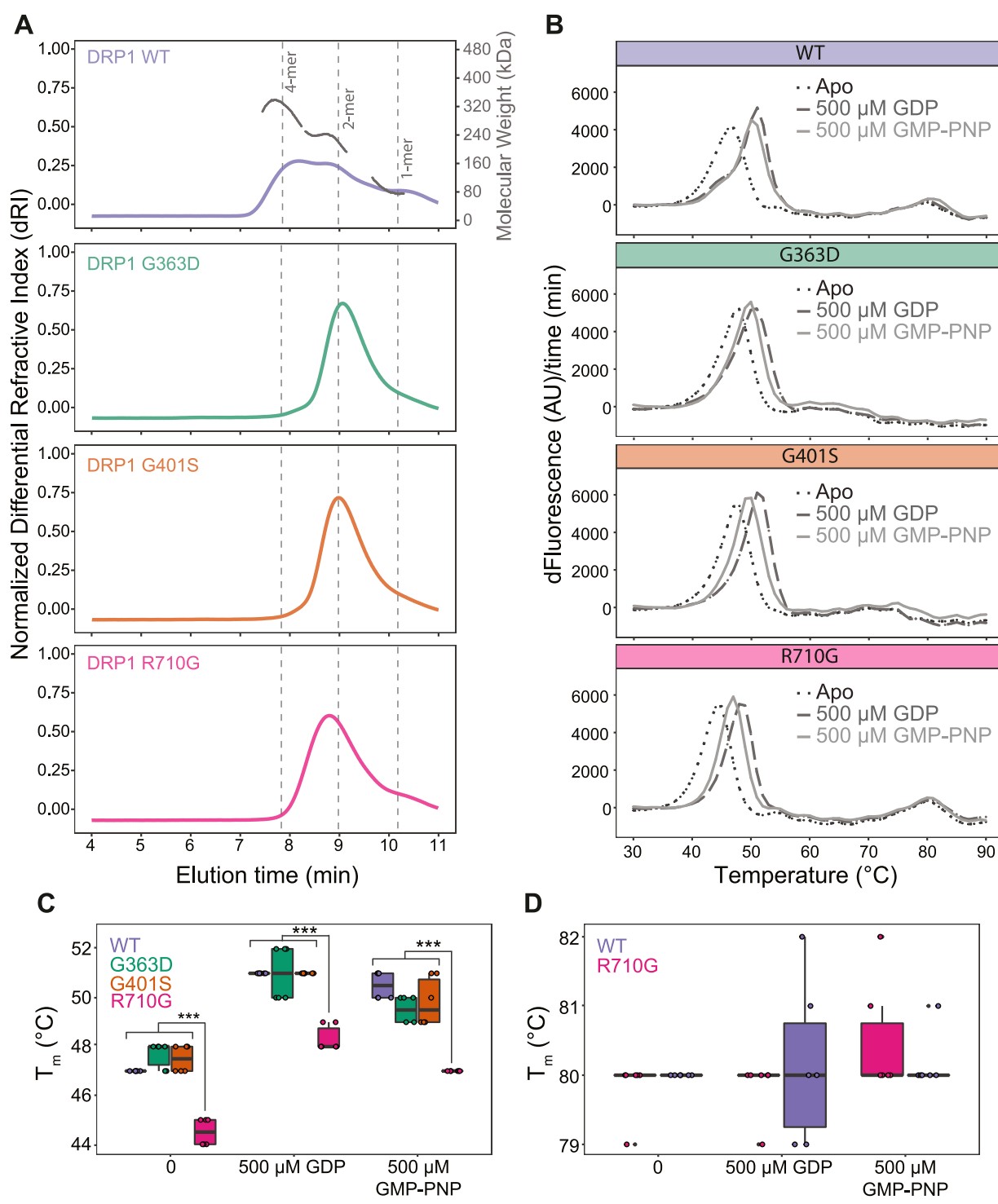

**Figure 6. Patient DRP1 variants alter DRP1 assembly-state and melting temperature.**
**(A)** SEC-MALS analysis of WT DRP1 (purple trace), DRP1 G363D (green trace), DRP1 G401S (orange trace), and DRP1 R710G (magenta trace) to assess for differences in multimeric distributions. Overlay of normalized differential refractive index of all protein samples (200 $\mu$g, 2.0 mg/ml) with peaks corresponding to monomeric, dimeric, and tetrameric oligomer species labelled as determined by predicted molecular masses of each multimeric species. Data normalized and scaled to allow for easier comparison because of slight differences in amount of protein loaded onto the column. **(B)** Melt curves of WT DRP1 and patient variants. Thermafluor analysis of protein unfolding of WT DRP1 (5.0 $\mu$M) and three patient variants (G363D, G401S, and R710G) either alone (black, dotted line), in the presence of 500 $\mu$M GDP (dark grey, dashed line), or GMP-PNP (light grey, solid line). Data plotted as the first derivative of the fluorescence signal with respect to time. **(C, D)** $T_m$ values determined from the temperature corresponding to the maximum fluorescence value in the absence of and presence of 500 $\mu$M GDP or GMP-PNP. **(C)** Thermafluor analysis of the first protein unfolding event reported as the melting temperature ($T_m$) of WT DRP1 (5.0 $\mu$M) and three patient variants either alone, or in the presence of 500 $\mu$M GDP or GMP-PNP. **(D)** Thermafluor analysis of the second protein unfolding event reported as the melting temperature ($T_m$). Only WT and R710G shown as they are the only two constructs

G363D, as well as other stalk domain variants including the lethal A395D substitution and G350D (Chang et al, 2010). Notably, R710G had an earlier peak elution time than G363D and G401S, 8.79 min versus 9.07 and 8.99 min, respectively. This suggests that R710G likely retains some ability to assemble into higher order oligomeric species, observed as a leftward shift in the peak elution time because of fast-exchange between dimeric and tetrameric species, contrary to G363D and G401S. In addition, treatment of total cell lysates derived from control and *DNM1L* patient fibroblasts with a chemical cross-linker BMH (bismaleimidohexane) resulted in increased formation of higher order oligomeric DRP1 complexes in P4 (Fig S7). Therefore, these data further support our SEC-MALS results, suggesting that the R710G variant retains more ability to assemble with wild-type DRP1 than other variants. Together, these results provide strong evidence that these disease-causing variants alter DRP1 ability to assemble, which is critical for mediating mitochondrial fission.

### Patient DRP1 variants are well-folded but have differing stability

Protein stability was evaluated using a fluorophore-based (SYPRO Orange) thermal shift assay and revealed the presence of two unfolding events in wild-type DRP1 (Fig 6B). Repeating the assay in the presence of either 500 $\mu$M GDP or 500 $\mu$M GMP-PNP showed increased stability of the first unfolding event upon nucleotide binding, but not the second. Therefore, we interpreted the first and second transitions as corresponding to the unfolding of the GTPase and stalk domains, respectively. Given the variable domain of DRP1 is intrinsically disordered (Strack & Cribbs, 2012; Fröhlich et al, 2013; Wenger et al, 2013; Rosdah et al, 2020; Mahajan et al, 2021), it is not surprising that a third unfolding event corresponding to this domain was not observed given no significant loss of secondary structure would be expected in this region upon unfolding. Both G363D and G401S were found to have only one distinct unfolding event corresponding to GTPase domain unfolding, consistent with the SEC-MALS data showing no higher order organisation. For wildtype, addition of GDP had little effect on the stalk domain transitions as expected (Fig 6C and D). By contrast, addition of GDP significantly increased the $T_m$ of the GTPase domain even more than GMP-PNP (Fig 6C and D), consistent with the known higher affinity of GDP over non-hydrolyzable GTP analogues for the GTPase domain (Fröhlich et al, 2013). This overall pattern was the same for all constructs indicating each variant is able to bind nucleotide, although DRP1 R710G showed a significantly lower GTPase domain melting temperature than WT, G363D and G401S, even in the presence of nucleotide, indicating that this variant destabilized the protein but not its ability to respond to nucleotide.

## Discussion

Here, we report the discovery of five patients with previously unreported variants in *DNM1L*, including only the second GED domain variant (p.Arg710Gly) to be identified to date. The p.Gly363Asp variant has previously been studied given its high degree of conservation across species, although this is the first report of a patient harbouring this pathogenic variant to our knowledge (Tanaka et al, 2006; Kobayashi et al, 2007; Chang et al, 2010; Otera et al, 2010; Kwapiszewska et al, 2019). The variants described here were predicted to be "likely pathogenic" according to ACGS guidelines, taking into account various criteria including variant allele frequency, functional studies, phenotypic fit and in silico predictions. In silico structural analysis of each variant concurred and predicted likely impairment of DRP1 oligomerisation (L230dup, G363D and G401S), GTP hydrolysis (L230dup and R710G) and protein stability (R710G) (Fröhlich et al, 2013; Kalia et al, 2018). Analysis of mitochondrial network morphology in fixed patient-derived cell lines revealed impaired mitochondrial fission leading to hyper-fused mitochondrial networks (Fig 3C) and in some cases enlarged mtDNA nucleoids (Fig S1), confirming dysfunctional DRP1 as the primary pathogenic factor in these patients. Furthermore, *DNM1L* variants present in P1 (p.Gly401Ser), P2 (p.Gly363Asp), P3 (p.Leu230dup) and P4 (p.Arg710Gly) also impaired normal peroxisomal fission (Fig 4A), which is not surprising given DRP1's prominent role in this process (Li & Gould, 2003; Koch et al, 2005; Tanaka et al, 2006; Kobayashi et al, 2007; Gandre-Babbe & Van Der Bliek, 2008; Otera et al, 2010; Koch & Brocard, 2012; Yamano et al, 2014). P1 (p.Gly401Ser), P2 (p.Gly363Asp) and P4 (p.Arg710Gly) *DNM1L* variants caused decreased DRP1 recruitment to the mitochondria (Fig 3D), but only P1 and P2 had decreased DRP1-peroxisome co-localisation (Fig 4B), suggesting that impaired DRP1 p.Arg710Gly peroxisomal fission occurs through a different mechanism. These data indicate that p.Arg710Gly mediated impairments are not simply due to a lack of DRP1 at the peroxisomal membrane, but may be due to impaired enzyme function with preservation of DRP1–peroxisome recruiter interactions, which are lost with the p.Gly363Asp and p.Gly401Ser variants.

To evaluate the effects of these mutations on DRP1, we performed a series of experiments designed to elucidate the specific mechanisms underpinning impaired function. The GTP hydrolysis activity is essential for DRP1 function. Interestingly, we found that G363D and G401S had increased GTP hydrolytic activity compared to WT DRP1, whereas R710G had decreased activity (Fig 5A). Previous studies examining the G363D variant have reported mixed hydrolysis results including no effect on hydrolytic activity (Clinton et al, 2016), or impaired hydrolysis (Tanaka et al, 2006; Chang et al, 2010). Given these discrepancies, it cannot be ruled out that differences in GTP hydrolysis may be due to variations in recombinant protein constructs or preparation methods (e.g., DRP1 isoforms, N- versus C-terminal tags, and calmodulin versus histidine purification tags) (Clinton et al, 2020). One might anticipate that increased GTP hydrolysis would result in increased fission intracellularly. However, it is possible that these results are representative of futile GTP cycling in which G363D and G401S retain hydrolytic capabilities but are unable to assemble into the higher order oligomeric species.

with a prominent second unfolding event. Data are representative of two independent experiments, each with three technical replicates. \*\*\*$P < 0.00001$. Differences between $T_m$ values of all constructs alone in comparison with constructs with 500 $\mu$M GDP or 500 $\mu$M GMP-PNP significant to $P < 0.003$. $T_m$ values of all constructs with 500 $\mu$M GDP in comparison to 500 $\mu$M GMP-PNP are significant to $P < 0.03$ except for R710G with 500 $\mu$M GDP in comparison to R710G with 500 $\mu$M GMP-PNP where $P < 0.0003$.

Both G363D and G401S appear to be in mutational hotspots (Fig 1B) with multiple variants in nearby regions reported including G350R, G362S, G362D, A395D, A395G, R403C, L406S, and E410K (Chang et al, 2010; Fahrner et al, 2016; Sheffer et al, 2016; Vanstone et al, 2016; Zaha et al, 2016; Ryan et al, 2018; Whitley et al, 2018; Vandeleur et al, 2019; Verrigni et al, 2019). These variants reside spatially close to each other within the stalk domain of the protein, a region important for mediating protein oligomerisation (Fröhlich et al, 2013; Francy et al, 2017), which in turn is critical for stabilization of DRP1–MFF complexes post recruitment to the mitochondria (Clinton et al, 2016) as well as assembly with MID49 (Kalia et al, 2018). This suggests the variants may have impaired fission secondary to diminished higher order assembly and/or poor recruitment to the mitochondria secondary to impaired DRP1–recruiter interactions. Consistent with this, both p.Gly363Asp and p.Gly401Ser have decreased DRP1 at the mitochondria as determined by DRP1-TOM20 co-localisation analysis (Fig 3D). Therefore, the decrease in mitochondrial fission despite increased GTP hydrolysis for both G363D and G401S likely stems from a lack of DRP1 recruitment and productive fission activity at the mitochondria.

In addition, our SEC-MALS data suggest that both G363D and G401S are unable to attain higher order species as they eluted in a primarily dimeric population (Fig 6A), consistent with previous reports on G363D (Chang et al, 2010; Clinton et al, 2016; Kwapiszewska et al, 2019). The glycine at position 401 is one of four highly conserved amino acids (GPRP, 401-404) located at the assembly interface where it is involved in mediating oligomerisation of proteins within the Dynamin superfamily including dynamin, DRP1, and MxA (Gao et al, 2010; Faelber et al, 2011; Ford et al, 2011). Like dynamin, these four residues required mutation to AAAA to prevent oligomerisation and inherent disorder of the loop region to achieve crystallisation of DRP1 (Fröhlich et al, 2013). Therefore, it is likely that both substitutions directly impair higher order assembly and may also disrupt local secondary structure given these variants did not exhibit a clear unfolding of the stalk domain by thermal shift analysis.

From a clinical mitochondrial disease perspective, it is interesting that both P1 (c.1201G>A, p.Gly401Ser) and P2 (c.1088G>A, p.Gly363Asp) exhibited cardiac complications, including end-stage dilated cardiomyopathy with previous signs of hypertrophic cardiomyopathy in P1. Of the previously reported variants, c.1228G>A, p.Glu410Lys is the only pathogenic human *DNM1L* variant that has been reported to result in severe cardiac involvement, which ultimately resulted in death of the patient at 8 mo of age (Vandeleur et al, 2019). Cardiac involvement in patients with *DNM1L*-related mitochondrial disease has previously been postulated because a C452F substitution in mouse DRP1 (position p.Cys446Phe in human DRP1 NP_036192.2) was shown to cause dilated cardiomyopathy (Cahill et al, 2015). Concordantly, a 3-mo-old patient who initially presented with infantile parkinsonism-like symptoms was identified to possess the same C446F substitution and died at 2.5 yr of age because of sudden cardiac arrest (Díez et al, 2017). However, no post-mortem evaluation was performed to determine the cause of cardiac arrest. It would therefore seem appropriate that patients with confirmed pathogenic *DNM1L* variants follow a cardiac surveillance programme, as is in place for other forms of mitochondrial disease, with a view to appropriate pre-emptive treatment.

In general, pathogenic variants involving the stalk domain of DRP1 also appear to be more severe than those affecting the GTPase domain which primarily present as optic abnormalities with or without concurrent neurological and developmental findings (Gerber et al, 2017; Hogarth et al, 2018; Whitley et al, 2018; Longo et al, 2020; Wei & Qian, 2021). We note a similar trend in our cohort with P1, P5 (c.1201G>A, p.Gly401Ser), and P2 (c.1088G>A, p.Gly363Asp) experiencing an earlier onset of more severe symptoms, faster disease progression, and early death, whereas P3 (c.687_689dupATT, p.Leu230dup) and P4 (c.2128A>G, p.Arg710Gly) had a later onset and lived to an older age. Of note, P3 (p.Leu230dup) and P4 (p.Arg710Gly) also exhibited less severe peroxisomal defects compared with P1 (p.Gly401Ser) and P2 (p.Gly363Asp). It may be that concurrent mitochondrial and peroxisomal defects lead to more severe phenotypes and disease progression. Consistent with this, several other non-lethal DRP1 variants, located primarily in the GTPase domain, resulted in cells with normal peroxisome morphology despite having impaired mitochondrial networks (Chao et al, 2016; Gerber et al, 2017; Whitley et al, 2018) (Table S1).

In true peroxisomal biogenesis disorders (PBDs), lipid metabolism, among other peroxisome-related metabolic pathways, is impaired. Clinically, *DNM1L* and PBD patients have phenotypic overlap including developmental delays, seizures, hypotonia, facial dysmorphism, and vision impairment. Unlike PBD patients though, *DNM1L* patients do not typically develop renal or hepatic dysfunction, skeletal abnormalities, or cataracts (Waterham & Ebberink, 2012). Given these similarities, and the peroxisome fission abnormalities in many *DNM1L* patients, one might hypothesize that *DNM1L* patients would display similar biochemical profiles, with elevated very long-chain and branched-chain fatty acids (De Biase et al, 2019). Unfortunately, there remains a dearth of *DNM1L* patient reports that analyse both peroxisomal morphology and perform the necessary analyses to fully evaluate peroxisomal function. Based on data currently available, there is not a clear correlation between laboratory findings, peroxisome morphology, and disease severity with some variants displaying normal peroxisome morphology with normal laboratory tests (p.Gly362Ser) (Sheffer et al, 2016), normal peroxisomes with elevated plasma VLCFA and normal pristanic acid (p.Gly32Ala) (Whitley et al, 2018), abnormal peroxisomes with normal laboratory tests (p.Ser36Gly, p.Glu116-Lysfs*6; p.Gly362Ser; p.Ile512Thr, p.Gly362Asp; p.Gly350Arg, and p.Tyr691Cys) (Chao et al, 2016; Nasca et al, 2016; Verrigni et al, 2019), and abnormal peroxisomes with abnormal laboratory tests (p.Ala395Asp) (Waterham et al, 2007). Several studies noted abnormal peroxisomal morphology but did not perform lipid profiling (Chao et al, 2016; Zaha et al, 2016; Longo et al, 2020), and it is unclear whether these patients may have had abnormal results (Table S1). Although traditional peroxisome functional tests may not be fruitful diagnostically, future studies using lipidomic approaches may capture more nuanced metabolic changes that occur, identifying potential biomarkers for *DNM1L*-associated disease with peroxisome involvement. Ultimately, *DNM1L* disorders appear to derive primarily from mitochondrial defects and the degree of peroxisome-driven pathology remains unclear, but likely secondary.

Unfortunately, we were unable to obtain full-length recombinant DRP1 L230dup (P3) for in vitro studies. Given this residue's relative proximity to the nucleotide-binding site, a duplication event is

likely to disrupt GTP binding. This would have direct impacts on GTP hydrolysis and resulting fission activity. DRP1 L230 is also near the DRP1–MID49 interface and the duplication may selectively inhibit recruiter interactions. Currently, only the structure of DRP1 in complex with MID49 has been solved (Kalia et al, 2018), so it is possible that other recruiting proteins bind at alternate locations enabling residual DRP1 activity to be performed. Alternatively, and contrasting a dominant negative mechanism, this allele is catalytically dead and residual DRP1 activity is maintained by the wild-type allele. In support of this, patient fibroblasts demonstrated a milder hyperfusion of mitochondrial reticula compared with the other variants and they lived to 20 yr of age, suggesting slower disease progression.

Intramolecular interaction between a monomer's GTPase Effector Domain (GED), the N-terminal GTPase domain, and stalk domain, as well as interactions between adjacent GEDs are essential for regulation of DRP1 GTP hydrolysis (Pitts et al, 2004; Zhu et al, 2004; Chang & Blackstone, 2007). This is a common feature in all dynamin proteins (Muhlberg et al, 1997; Schumacher & Staeheli, 1998; Di Paolo et al, 1999; Sever et al, 1999; Shin et al, 1999; Smirnova et al, 1999; Zhang & Hinshaw, 2001) where removal of the GED in dynamin or DRP1 does not prevent nucleotide binding or higher order assembly but decreases GTPase activity (Muhlberg et al, 1997; Zhu et al, 2004). Similarly, R710G can still bind GTP, evidenced by its ability to hydrolyse GTP and stabilisation of the GTPase domain upon nucleotide binding but has decreased GTPase activity. Mutation of R725 in dynamin (R710 in human DRP1 [NP_036192.2] and both located in the hinge 1 region) prevents stimulation of GTPase activity by the GED domain, suggesting it is a key residue involved in sensing and transmitting assembly information to the GTPase domain (Sever et al, 1999). The hinge 1 region has also been shown to be important for MxA function which shares structural properties with the family of dynamin-like GTPases. However, disruption of MxA R640 or E632 (equivalent to R710 and E676 in human DRP1 [NP_036192.2]) impairs higher order oligomerisation and decreased the off-rate of GTP, thus causing increased GTP hydrolysis which is opposite of what is observed in dynamin (Sever et al, 1999; Gao et al, 2011). Nearby dynamin residue K694 (equivalent human DRP1 residue: K679) is also located in the GED, but mutation results in impaired assembly, suggesting it lays at the interface between adjacent GEDs where it stabilizes their interaction during assembly (Sever et al, 1999). A previously reported de novo p.Tyr691Cys DRP1 variant in the fifth α-helix of the stalk portion of the GED was proposed to disrupt GED–GTPase interactions (Batzir et al, 2019), but it seems more likely that this substitution would negatively impact GED–GED assembly given its location at this interface. Interestingly, the c.2072A>G, p.Tyr691Cys DNM1L patient, and our c.2128A>G, p.Arg710Gly (P4) had similar, less severe phenotypes compared with stalk domain variants and presented with epilepsy, optic atrophy, impaired mobility, and prominent cyclical vomiting.

Therefore, we predict that R710G is pathogenic because of a disruption in the sensing mechanism that facilitates assembly-driven increases in GTP hydrolysis. Furthermore, this variant had the greatest loss of recruitment to the mitochondria in patient fibroblasts, suggesting this process, or region of the protein, may be important for proper DRP1–mitochondrial recruiter recognition. It is unclear if the substitution results in direct disruption of GED–GTPase domain interaction, or if it is a downstream mechanism. Supporting a direct disruption, R710G results in a lower $T_m$ for the GTPase domain, albeit with retained nucleotide-binding capabilities, reflective of decreased protein stability, possibly due to loss of the intramolecular GED–GTPase domain interactions. It is therefore not surprising that this patient had lower protein levels of DRP1, and this may be reflective of increased protein degradation secondary to the decreased stability, whereas the other patients did not, suggesting haploinsufficiency is not a major driver of pathology in those cases, which has been noted for other variants as well (Whitley et al, 2018). R710G is perhaps somewhat assembly deficient compared with wildtype, but more assembled than G363D or G401S and is found in a dynamic equilibrium between a dimeric and tetrameric state.

There are nine known DRP1 isoforms that arise from differential splicing in the GTPase or variable domains, with isoforms differing based on their inclusion, or lack of, a 13–amino acid insert in the GTPase domain (A insert) and a partial or full 37–amino acid insert in the variable domain (B insert) (Rosdah et al, 2020). These isoforms have varying GTPase rates in the presence of cardiolipin, a primary component of the mitochondrial outer membrane, or in response to the DRP1 recruiter MFF (Macdonald et al, 2016). Currently, none of the reported variants are found within the A or B insert, suggesting all DRP1 isoforms in patients would be affected. This raises the question of why neuronal tissue is predominantly affected in this patient population. It may be that certain isoforms are more tolerant of substitutions, experiencing fewer or less severe impacts on protein oligomerisation or GTP hydrolysis. Genetic mosaicism may also play a role in patients with milder, or perhaps even subclinical phenotypes. It is also unclear why fetal development is grossly normal, given the preponderance of heterozygous dominance among DNM1L variants. A role for DRP1 in development is still emerging, but evidence supports the importance of DRP1 as global knockout is embryonic lethal in mouse models (Ishihara et al, 2009; Wakabayashi et al, 2009).

Here, we have described with mechanistic precision how pathogenic variants disrupt DRP1 biophysical activity and lead to mitochondrial hyperfusion. We document that divergent mechanisms including combinations of aberrant stability, organellar recruitment, assembly, and GTPase activity contribute to pathogenesis caused by mutations in different domains of DRP1. In summary, a thorough understanding of how DRP1 function is impaired in human disease will provide insight into the diverse phenotypes and variable disease severity associated with pathogenic DNM1L variants. A systematic characterisation of patient presentation and progression will assist in the timely identification of other patients with rare DNM1L variants, whereas understanding the specific molecular mechanisms underlying DRP1 function will promote the development of targeted therapeutics with a goal of restoring mitochondrial fission to non-pathological levels. Crucially, our work details the first example of a patient with a DNM1L variant in the hinge region which will be crucial to answering an outstanding question: how assembly information is transmitted to the GTPase domain to stimulate GTP hydrolysis in the dynamin superfamily.

# Materials and Methods

## Ethical statement

Written informed consent for diagnostic molecular genetic analysis and research-based studies was obtained from all patients in accordance with the Declaration of Helsinki protocols and ethical approvals of local institutional review boards.

## Diagnostic studies of skeletal muscle biopsies

Available skeletal muscle biopsies were subjected to routine diagnostic investigations, including diagnostic TEM studies of muscle from P1. Diagnostic in vitro spectrophotometric measurements of respiratory chain complex activities were undertaken in P2, P3, and P4 muscle according to standard procedures (Kirby et al, 2007). Complex I and IV–immunodeficient muscle fibres in P2 were determined by a quadruple fluorescent IHC assay of OXPHOS function, which evaluates protein levels of mitochondrial subunits of complex I (NDUFB8) and complex IV (COX1). In addition to the immunofluorescence labelling of muscle sections using antibodies against the above described OXPHOS complexes, the mitochondrial mass was quantified using an antibody against the outer mitochondrial membrane protein—porin (VDAC) and the myofibre boundaries were labelled with anti-laminin, a membrane glycoprotein as previously described (Rocha et al, 2015).

## Molecular genetics studies

All patients underwent routine mtDNA diagnostic testing that excluded variants in the mitochondrial genome. Next generation sequencing strategies followed by filtering and candidate variant analysis were undertaken to elucidate the molecular bases of studies on mitochondrial disease patients. GnomAD (https://gnomad.broadinstitute.org/) database was used for minor allele frequency analysis (≤0.01%). In silico pathogenicity tools were used to assess the pathogenicity of candidate variants and classified as "likely pathogenic" using the Association of Clinical Genomic Science (ACGS) and The American College of Medical Genetics and Genomics (ACMG) guidelines (Richards et al, 2015) (https://www.acgs.uk.com/media/11631/uk-practice-guidelines-for-variant-classification-v4-01-2020.pdf).

Family trio WES analysis was performed on P1 using the Agilent Sure Select Human All Exon Kit v6 according to the manufacturer's instructions, followed by sequencing on an Illumina NextSeq platform. For P2, targeted NGS sequencing using a custom 84.38-Kb Ampliseq panel (Life Technologies) was initially performed to capture relevant regions of 50 Complex I genes as previously described (Alston et al, 2016). Sequencing was performed using the Ion PGM 200 Sequencing Kit on an Ion Torrent PGM Sequencer. Variant calling was undertaken using the proprietary Ion Torrent Variant Caller plugin and sequence variants were annotated using wANNOVAR for prioritisation and classification. Further to targeted NGS, trio WES analysis was performed using Agilent SureSelectXT All Exon v.5 according to the manufacturer's instructions, followed by sequencing on an Illumina HiSeq2500 platform and in-house

pipelines were used for variant calls as previously described (Taylor et al, 2014; Rocha et al, 2015). For P3, whole genome sequencing was performed by Genomics England via the 100,000 genomes project. WES and variant filtering and prioritisation was performed in P4 as previously described (Taylor et al, 2014; Thompson et al, 2016). WES analysis was also performed on P5 as described in P1 using the Agilent Sure Select Human All Exon Kit v6 and sequencing on an Illumina NextSeq platform.

## In silico analysis and structural modelling

The structures of DRP1 (PDB: 4BEJ), DRP1 in complex with GDP.AlF$_4$ (3W6P), DRP1 in complex with GMP-PCP (3W6O), and co-assembled DRP1-MID49 (PDB: 5WP9) were used to assess the structural implications of the patient mutations using PyMOL by Schrödinger (https://pymol.org/2/). In silico mutagenesis was performed using Modeler software with standard parameters (https://salilab.org/modeller/).

## Cell lines

Primary patient fibroblasts and age-matched controls were grown in high-glucose Dulbecco's Modified Eagle Medium (Gibco) supplemented with 10% (vol/vol) FBS, 1% non-essential amino acids, 1.0 mM sodium pyruvate, 50 μg/ml uridine, 50 U/ml penicillin, and 50 μg/ml streptomycin at 37°C in an atmosphere of 5.0% CO$_2$. All primary control and patient fibroblasts used in this study were under P0+12 passages.

## Mitochondrial network analysis using the Mitochondrial Analysis (MiNa)

Asynchronized control and patient fibroblasts were cultured overnight on 35-mm Ibidi m-dishes (Ibidi, 88156) before incubation in 0.15% PicoGreen (P7581; Invitrogen) for 30 min, then washed and incubated in 5.0 nM TMRM (T668; Invitrogen) for 30 min. Z-stack images were taken on a VisiTech iSIM with a 100× objective before processing using Fiji to generate maximum projection images. These images were analysed using the Mitochondrial Analysis (MiNa) tool on Fiji and for each image the mean mitochondrial network length calculated by multiplying the mean branch length by the mean number of branches per network. Statistically significant differences were calculated via non-parametric one-way ANOVA and Dunn's multiple comparisons using GraphPad Prism.

## Analysis of mtDNA nucleoids, mitochondrial network, and peroxisomal morphology using the Columbus system

Cells were synchronised overnight by starvation using DMEM medium with 0.1% FBS (Mitra et al, 2009). G0-arrested cells were plated out on to 96-well plates and cultured in DMEM containing 10% FBS for 24 h before incubation with 0.15% PicoGreen (P7581; Invitrogen) for 30 min at 37°C. After three washes in Fluorobrite DMEM (A1896701; Thermo Fisher Scientific), Z-stack images were taken on a Zeiss CellDiscoverer7 microscope with 50× water objective (NA 1.2). Maximum projection images were analysed using the Columbus (PerkinElmer) software system and for each image field, the

proportion of enlarged nucleoids classed as over 1.5 $\mu m^2$ were calculated in the total nucleoid pool. Statistically significant differences were calculated via non-parametric one-way ANOVA using GraphPad Prism.

For immunofluorescence analysis cells were synchronised as described above and fixed with 4% paraformaldehyde in 1× PBS for 15 min at 37°C. Following three washes in PBS, cells were incubated for 10 min with 50 mM $NH_4Cl$ to quench the paraformaldehyde, washed three times in PBS, and permeabilized with 0.1% Triton X-100 in PBS for 10 min at RT. Subsequently, cells were washed three times with PBS, blocked with 5% FBS in PBS for 10 min at RT, and incubated with primary antibodies in diluted blocking buffer overnight at 4°C. Primary antibodies (anti-DRP1 BD Biosciences #611113 [1:500 dilution], anti-TOM20 Santa Cruz sc-17764 [1:500 dilution], anti-TOM20 Abcam ab186735 [1:2,000 dilution], and anti-PMP70 Abcam ab3421 [1:3,000 dilution]) were washed off with PBS (3 × 5 min) and appropriate secondary antibodies conjugated to Alexa Fluor 488 or 647 (1:1,000 dilution Molecular Probes; Invitrogen) and Hoechst 33342 (Thermo Fisher Scientific) stain (1:5,000 dilution) were applied for 1 h at RT. After 3 × 5 min washes with PBS, cells were analysed using a Zeiss CellDiscoverer7 microscope. Z-stack images were taken with a 50× water objective (NA 1.2) before maximum-intensity projections were analysed using Columbus (PerkinElmer) for mitochondrial and peroxisomal network length and mtDNA nucleoids size. Statistically significant differences in mitochondrial network length were calculated via non-parametric one-way ANOVA using GraphPad Prism.

### Immunofluorescence DRP1 co-localisation studies

For DRP1 co-localisation studies, synchronised cells were labelled with antibodies against DRP1, TOM20, and PMP70 as described above. Images were prepared for co-localisation analysis in Fiji (ImageJ) using two separate ImageJ macros: one to split channels into separate folders and one to generate stacks of Z-projections for each channel as well as a merged max-intensity projection image of all three channels (Schindelin et al, 2012) (https://github.com/Hill-Lab/DNM1L-Variants-Scripts). For the cells immunostained with anti-PMP70, the same ImageJ macros were used but included a separate rotation (1–2°) and crop step to correct for slight skewing of the stitched images. Cells were outlined to create regions of interest (ROIs) using the software CellProfiler (McQuin et al, 2018). Single channel maximum-intensity projection images were corrected for illumination variations and primary objects were classified as nuclei using adaptive Otsu thresholding on the DAPI channel. Secondary objects were classified as cells using the DRP1 channel with nuclei as the input objects. For cells co-stained with anti-TOM20, cells were identified using the Watershed-Image feature of CellProfiler with the Global Minimum Cross-Entropy thresholding method. Cell outlines for cells co-stained with PMP70 were created using the same method except no illumination variation correction was performed and cells were identified using the adaptive Otsu thresholding method.

Cell outlines were exported as a .png image file and used as the ROIs for co-localisation analysis in ImageJ. Cell outlines were visually inspected and cells that were not adequately outlined were corrected manually in ImageJ. The ImageJ coloc2 plugin was then used to calculate the Pearson's Correlation between endogenous DRP1 and either endogenous TOM20 or endogenous PMP70 using the selected ROI regions from the maximum intensity projection images. RStudio (1.4.1106) (RStudio Team, 2021) was used to tidy and compile this data using tidyverse 1.3.0 (Wickham, 2017), plot as box plots using ggplot2 (3.3.3) (Wickham, 2016), and perform one-way ANOVA with post-hoc Tukey's Honest Significant Difference test (https://github.com/Hill-Lab/DNM1L-Variants-Scripts).

### Protein expression and purification

Recombinant DRP1 isoform 1 was expressed using a pET29b+ vector as a $DRP1^{1–736}$-$His_6$ fusion protein in BL21(DE3) *Escherichia coli* as previously described (Cahill et al, 2015; Bordt et al, 2017). Transformed cells were grown at 37°C in Luria broth containing kanamycin (30 g/ml) to an $OD_{600}$ of ~1.0 with 0.5 mM isopropyl 1-thio-b-D-galactopyranoside (IPTG). After 16–18 h, cells were harvested by centrifugation and resuspended in Buffer A (20 mM Hepes, pH 7.4, 500 mM NaCl, 40 mM imidazole, and 0.02% sodium azide) containing protease inhibitors (Roche Applied Science). Cells were lysed with an EmulsiFlex C3 homogenizer (Avestin) at 15,000 p.s.i. and protein lysate was clarified through centrifugation at 15,000 rpm for 45 min at 4°C using a JA–20 fixed-angle rotor in a Beckman J2–21 centrifuge. Clarified lysate was applied to a nickel affinity column (Sepharose high performance beads; GE Healthcare) equilibrated in Buffer A using an FPLC. The column was washed with 10 column volumes each of buffers B (20 mM Hepes, pH 7.4, 500 mM NaCl, 40 mM imidazole, 10 mM KCl, 1.0 mM ATP, and 0.02% sodium azide) and C (20 mM Hepes, pH 7.4, 500 mM NaCl, 40 mM imidazole, 0.5% [wt/vol] CHAPS, and 0.02% sodium azide). Protein was eluted with Buffer D (20 mM Hepes, pH 7.4, 500 mM NaCl, 500 mM imidazole, and 0.02% sodium azide) and peak fractions were pooled, concentrated to ~1.0–2.0 ml using Vivaspin 20 centrifugal concentrators (GE Healthcare) with a molecular weight cut-off of 50 kD, and dialyzed overnight at 4°C in GTPase reaction buffer (20 mM Hepes, pH 7.4, 150 mM KCl, 2.0 mM $MgCl_2$, 1.0 mM DTT, 0.5 mM EDTA, and 0.02% sodium azide). Protein concentration was determined by measuring absorbance at 280 nm in the presence of 6.0 M guanidine HCl with a theoretical extinction coefficient of 35,870.96. Protein was then flash frozen in liquid nitrogen in single use 100–200 $\mu l$ aliquots and stored at −80°C. All studied DRP1 variants were obtained through Quickchange mutagenesis (Stratagene) with a pET29b+−$DRP1^{1–736}$ (isoform 1) construct (primers available upon request). DRP1 variants were induced with 0.25 mM IPTG and otherwise expressed and purified akin to DRP1–WT.

### GTPase activity measurements

DRP1 GTPase activity was measured using a continuous, regenerative coupled GTPase assay which reports on GTP hydrolysis that is directly proportional to the depletion of NADH (Ingerman & Nunnari, 2005). Depletion of NADH was measured at $Abs_{340}$ for 45 min at 25°C using a Molecular Devices FlexStation 3 Multi–Detection Reader with Integrated Fluid Transfer. Reactions (150 $\mu l$) of 1.0 $\mu M$ DRP1 (WT or variant) and 150 mM NaCl were performed in a flat–bottom 96-well plate (Corning Costar) in GTPase reaction buffer (25 mM Hepes, pH 7.4, 10 mM $MgCl_2$, 1.0 mM phosphoenolpyruvate (PEP),

7.5 mM KCl, 0.8 mM NADH, and 20 U/ml pyruvate kinase/lactate dehydrogenase) at the following GTP concentrations: 0, 10, 30, 70, 100, 300, 500, 700, 1,000, 1,300, 1,700, and 2,000 $\mu$M. Reactions were started by addition of 10 $\mu$l of 15× concentrated GTP stocks to each well. Data were imported into RStudio (1.4.1106) (RStudio Team, 2021) using readxl (1.3.1) (Wickham & Brian, 2019) and tidied using tidyverse (1.3.0) (Wickham, 2017). Depletion of NADH at $Abs_{340}$ was converted to GTPase activity rates using Equation (1) and kinetic parameters were determined through global fitting of the data to a Michaelis–Menten model (Equation (2)). $k_{cat}$ values were then determined using Equation (3). Activity from each DRP1 variant was collected from three independent preparations and reported as means ± SD. Statistical significance was determined by a one-way ANOVA followed by Tukey's honest significant difference test. Plots were generated in RStudio using ggplot2 (3.3.3) (Wickham, 2016), gridExtra (2.3) (Baptiste, 2017), and RColorBrewer (1.1-2) (Neuwirth, 2014) (https://github.com/Hill-Lab/DNM1L-Variants-Scripts).

$$\begin{aligned} &\text{GTPase activity } (\text{min}^{-1}) \\ &= \frac{\Delta Abs340}{min} \bigg/ \left(\frac{6220}{Mcm} \times p \bigg/ \frac{1e6\mu M}{M}\right) \bigg/ \big[DRP1\big], \end{aligned} \quad (1)$$

where $\Delta Abs340$ = change in Abs at 340 nm for the steady-state linear depletion, Vol = volume of reaction, which is 150 $\mu$l here, 6,220/Mcm = extinction coefficient of NADH, [DRP1] = concentration of DRP1 used in assay, which was 1.0 $\mu$M unless otherwise noted, and p = path length of well, which was determined to be 0.4649 cm in our assay setup.

$$V_0 = \frac{V_{max} \times [GTP]}{[GTP] + K_{0.5}}, \quad (2)$$

where $V_0$ = initial velocity of reaction, $V_{max}$ = maximal velocity of reaction, [GTP] = concentration of GTP (i.e., substrate), and $K_{0.5}$ = value of [GTP] at $V_0$ = 0.5 × $V_{max}$ and is a generalized Michaelis–Menten constant.

$$k_{cat} = \frac{Vmax}{[DRP1]}, \quad (3)$$

where $V_{max}$ = maximal velocity of reaction, [DRP1] = concentration of DRP1 used in assay, which was 1.0 $\mu$M.

### Size-exclusion chromatography with multiangle laser light scattering and differential refractive index

Wild-type and variant DRP1[1-736]-His[6] fusion proteins were purified as described above. Aliquots (400 $\mu$L total volume) were thawed on ice and dialyzed overnight at 4°C into column running buffer (20 mM Hepes, pH 7.4, 150 mM KCl, 2.0 mM MgCl$_2$, 0.5 mM EDTA, 1.0 mM DTT, and 0.02% sodium azide filtered through a 0.02-micron filter using vacuum filtration). Protein concentrations were determined as stated above following dialysis. Samples were injected (100 $\mu$L of 2.0 mg/ml) and chromatographed at 1.0 ml/min at 25°C on a BioSep HPLC size-exclusion column (BioSep-SEC-S 4000, 300 × 7.8 mm) equilibrated with column running buffer with a guard column (08543-TSKgel Guard SWXL, 6.0 mm ID × 4.0 cm, 7.0 $\mu$M) in place. The eluate was detected using a DAWN-EOS multiangle laser light

scattering instrument and the Optilab refractive index detector (Wyatt Technologies). Data analysis was performed using the ASTRA software package (Wyatt Technologies). Data were imported into RStudio (1.4.1106) (RStudio Team, 2021) and tidied as described above. Traces were then normalized and centre-scaled to allow for easier comparison using caret (6.0-86) (Kuhn, 2020). Chromatograms were then generated using ggplot2 (3.3.3) (Wickham, 2016) and RColorBrewer (1.1-2) (Neuwirth, 2014) and plotted with molar mass (right axis) and normalized and scaled dRI (left axis) as a function of time (x-axis) (https://github.com/Hill-Lab/DNM1L-Variants-Scripts). Chromatograms are representative of two independent protein preparations for wild-type and each variant.

### Thermal shift assay

Reactions (30 $\mu$l) consisting of 5.0 $\mu$M DRP1 (WT or variant) and 5× SYPRO orange (excitation 470 nm/emission 570 nm) in DRP1 GTPase reaction buffer (20 mM Hepes, pH 7.4, 150 mM KCl, 2.0 mM MgCl$_2$, 1.0 mM DTT, 0.5 mM EDTA, and 0.02% sodium azide) ± 500 $\mu$M GDP or 500 $\mu$M GMP-PNP were set up in a 0.1-ml × 96-well white non-skirted PCR plate (PR1MA PR-PCR1196-W). Both GDP and GMP-PNP stock solutions were prepared in DRP1 GTPase reaction buffer (20 mM Hepes, pH 7.4, 150 mM KCl, 2.0 mM MgCl$_2$, 1.0 mM DTT, 0.5 mM EDTA, and 0.02% sodium azide) to the target concentration. PCR plates were heat treated at 95°C for 30 min to prevent SYPRO orange from interacting with polyethylene in plate giving erroneous fluorescence readings at 57°C. Protocol adapted from Huynh and Partch (Huynh & Partch, 2015). SYPRO orange fluorescence was measured in 1-min intervals with a temperature ramp of 1°C per minute using a Stratagene Thermoler Mx3005P. Data were imported into RStudio (1.4.1106) (RStudio Team, 2021) and tidied as described above. Melting temperatures ($T_m$) were determined by the temperature corresponding to the maximum value of the first derivative of the fluorescence signal. Statistical significance of $T_m$ value alterations were determined by a one-way ANOVA followed by Tukey's honest significant difference test. Plots were generated using ggplot2 (3.3.3) (Wickham, 2016) and RColorBrewer (1.1-2) (Neuwirth, 2014) with dFluorescence/dTime (y-axis) as a function of time (x-axis). Two biological replicates, each with three technical replicates, were used for analysis and plot generation (https://github.com/Hill-Lab/DNM1L-Variants-Scripts).

### Circular dichroism

Far UV circular dichroism was performed on a Jasco J-1500 CD Spectrometer with 0.05 mg/ml DRP1 (WT and variant) at 20°C using a 10-mm path length and 5 accumulation average. A continuous scanning mode at 100 nm/min with 3.0 nm bandwidth, 0.1 nm data interval from 190 to 300 nm was used. All samples were brought to an equivalent concentration in DRP1 GTPase reaction buffer (20 mM Hepes, pH 7.4, 150 mM KCl, 2.0 mM MgCl$_2$, 1.0 mM DTT, 0.5 mM EDTA, and 0.02% sodium azide) and then diluted to their final concentration in double-distilled 0.45-$\mu$m filtered H$_2$O to ensure equivalent concentrations of buffer components. Reference scans performed on each sample's empty cuvette, as well as a buffer-only sample, were subtracted from the final signal to remove background ellipticity due to residual buffer components. Molar

ellipticity was converted to mean residue ellipticity (Equation (4)) and data were scaled to a baseline of 0 at 260 nm using Microsoft Excel. Data were imported into RStudio (1.4.1106) (RStudio Team, 2021) and tidied as described above. Plots were generated using ggplot2 (3.3.3) (Wickham, 2016) and RColorBrewer (1.1-2) (Neuwirth, 2014) with CD signal in terms of mean residue ellipticity on the vertical axis and wavelength on the horizontal axis (https://github.com/Hill-Lab/DNM1L-Variants-Scripts). Protein concentrations were determined again after data collection using the absorbance and theoretical extinction coefficients at 205 and 214 nm to ensure that equivalent amounts of protein were used.

$$\Omega = \frac{\theta \times MRW}{10 \times c \times d} \qquad (4)$$

where $\Omega$ = mean residue ellipticity, MRW = mean residue weight calculated as the protein molecular weight/(N-1) where N = total number of residues, c = concentration (mg/ml), and d = path length (cm).

## Data Availability

All R scripts used for data analysis and visualization are available upon request and/or for download at https://github.com/Hill-Lab/DNM1L-Variants-Scripts.

## Supplementary Information

## Acknowledgements

R McFarland and RW Taylor are supported by the Wellcome Centre for Mitochondrial Research (203105/Z/16/Z), Mitochondrial Disease Patient Cohort (UK) (G0800674), the Medical Research Council (MRC) International Centre for Genomic Medicine in Neuromuscular Disease (MR/S005021/1), the UK NIHR Biomedical Research Centre for Ageing and Age-related disease award to the Newcastle upon Tyne Foundation Hospitals NHS Trust, and the UK NHS Highly Specialised Service for Rare Mitochondrial Disorders of Adults and Children. M Oláhová and RW Taylor receive funding from the Pathology Society. R McFarland, M Oláhová, and RW Taylor receive funding from the Lily Foundation. OM Russell is supported by the Wellcome Centre for Mitochondrial Research (203105/Z/16/Z). CL Alston is supported by a National Institute for Health Research (NIHR) Post-Doctoral Fellowship (PDF-2018-11-ST2-021). This project was also supported by the following National Institutes of Health grants: TL1TR001437 and T32GM080202 (to KA Nolden), R01GM067180 (to RB Hill), and R01HL128240 (to ME Widlansky). The views expressed in this publication are those of the author(s) and not necessarily those of the National Health Service (NHS), the National Institute for Health Research (NIHR) or the Department of Health and Social Care. The content is solely the responsibility of the author(s) and does not necessarily represent the official views of the NIH. We would like to thank Dr. Julien Prudent from the MRC Mitochondrial Biology Unit at University of Cambridge for constructive discussions about the project. We also thank Laura Bone for technical support and the Newcastle University Bioimaging Unit for assistance with the microscopy. We thank the Exeter Genomics Laboratory and the Gastroenterology team at Sheffield Children's Hospital for their support.

## Author Contributions

KA Nolden: conceptualization, data curation, software, formal analysis, investigation, methodology, and writing—original draft, review, and editing.

JM Egner: conceptualization, data curation, formal analysis, investigation, methodology, and writing—original draft.

JJ Collier: data curation, software, formal analysis, investigation, methodology, and writing—original draft.

OM Russell: software, formal analysis, investigation, and writing—review and editing.

CL Alston: formal analysis and investigation.

MC Harwig: software, formal analysis, investigation, and writing—review and editing.

ME Widlansky: investigation.

S Sasorith: software and formal analysis.

IA Barbosa: formal analysis and investigation.

AGL Douglas: investigation.

J Baptista: formal analysis and investigation.

M Walker: investigation.

DE Donnelly: investigation.

AA Morris: investigation.

HJ Tan: formal analysis and investigation.

MA Kurian: investigation.

K Gorman: formal analysis and investigation.

S Mordekar: investigation.

C Deshpande: investigation.

R Samanta: investigation.

R McFarland: investigation and writing—review and editing.

RB Hill: supervision, funding acquisition, investigation, and writing—review and editing.

RW Taylor: data curation, supervision, funding acquisition, investigation, and writing—review and editing.

M Oláhová: conceptualization, data curation, formal analysis, supervision, funding acquisition, investigation, methodology, and writing—original draft, review, and editing.

### Conflict of Interest Statement

The authors have declared that no conflict of interest exists. RB Hill and KA Nolden have a financial interest in Cytegen, a company developing therapies to improve mitochondrial function. However, neither the research described herein was supported by Cytegen, nor was is in collaboration with the company.

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
