## [Reviewer comments · Life Science Alliance]

Life Science Alliance

Novel DNM1L variants impair mitochondrial dynamics through divergent mechanisms.

Monika Oláhová, Kelsey Nolden, John Egnér, Jack Collier, Oliver Russell, Charlotte Alston, Megan Harwig, Michael Widlansky, Souphatta Sasorith, Inês Barbosa, Andrew Douglas, Julia Baptista, Mark Walker, Deirdre Donnelly, Andrew Morris, Hui Jeen Tan, Manju Kurian, Kathleen Gorman, Santosh Mordekar, Charu Deshpande, Rajib Samanta, Robert McFarland, R. Hill, and Robert Taylor

DOI: <https://doi.org/10.26508/lsa.202101284>

Corresponding author(s): Monika Oláhová, Newcastle University

Review Timeline:

Submission Date:	2021-11-01
Editorial Decision:	2021-12-09
Revision Received:	2022-06-01
Editorial Decision:	2022-06-29
Revision Received:	2022-07-07
Accepted:	2022-07-07

Transaction Report:

December 9, 2021

Re: Life Science Alliance manuscript #LSA-2021-01284-T

Dr. Monika Oláhová
Newcastle University
Wellcome Centre for Mitochondrial Research
Framlington Place
Newcastle upon Tyne NE2 4HH
UNITED KINGDOM

Dear Dr. Oláhová,

Thank you for submitting your manuscript entitled "Novel DNM1L variants impair mitochondrial dynamics through divergent mechanisms." to Life Science Alliance. The manuscript was assessed by expert reviewers, whose comments are appended to this letter. We invite you to submit a revised manuscript addressing the Reviewer comments.

Thank you for this interesting contribution to Life Science Alliance. We are looking forward to receiving your revised manuscript.

Sincerely,

B. MANUSCRIPT ORGANIZATION AND FORMATTING:

Reviewer #1 (Comments to the Authors (Required)):

The authors report an interesting analysis of four newly identified putatively pathogenic variants in DNM1L in five unrelated patients suffering from early-onset multi-systemic mitochondrial disease and make note of a peculiar manifestation of cardiac pathology, which is rather uncommon in patients carrying mutations in this gene. This synergistic study between structural biologists (US) and mitochondrial geneticists (UK) has the potential to be of great value for increasing our understanding of DRP1 biology in health and disease. They present extensive clinical background, as well as the data obtained from whole genome sequencing to identify heterozygous mutations that compromise DRP1 function, notably identifying the first patient with a pathogenic variant in the hinge region of the protein. They use in silico modeling of the different variants in order to understand the effect of those nucleic acid change on the protein 3D structure and function. Using recombinant proteins carrying most of these variants, they assess the impact these variants in DRP1 have on its oligomerization capacity and GTPase activity. Finally, they assess mitochondrial morphology, mtDNA nucleoid organization, and OXPHOS protein levels in primary fibroblasts from these patients.

Their results bring new insights on the role of single nucleotide variants in mitochondrial dynamic and extend the list of disease-causing variants in this gene (which were nicely summarized in the supplemental table). The genetic, genomic, and structural data in particular are particularly convincing given their quantitative nature and also well presented, leaving little room for criticism (although I am not an expert in structural biology) and the interpretations of their data are, for the most part, parsimonious. Nevertheless, there are several points to address regarding the experiments performed on human cells carrying so-called pathogenic variants that must be addressed to satisfy the claims made in their study before this study is suitable for publication. These studies are essential to prove the pathogenic nature of these variants and may help clarify the surprising findings made in vitro. It is important that they be performed in quantitative fashion, which is lacking in all experiments (main and supplemental figures) performed on cultured cells with the exception of Figure 1A.

Major comments:

1. The authors state that "As expected, all four patient cell lines (P1-P4) demonstrated hyperfusion of mitochondrial networks relative to control cells, which was generally supported by quantification". However, in Figure 3A, for P3, the quantification shows no difference at all with C3 and C4, leading the reader to question this conclusion. Is this because the experiment was only performed twice with only >20 fields of view? Why so few cells? It appears as though some cells in Figure 3 are in the process of mitosis, which is already known to modulate mitochondrial morphology. Have these cells been synchronized? Cell cycle is proven to modulate mitochondrial morphology and this point should be addressed. At the very least, they should be excluded from the analysis and not used as representative images.
2. If the purported pathogenic variants negatively impact DRP1 function, cells carrying these variants should have blocked mitochondrial fission and blocked peroxisomal fission. The latter should be measured, especially since the original study describing a severe clinical outcome caused by DRP1 deficiency (Waterham 2007) exhibited both peroxisomal and mitochondrial morphology defects in fibroblasts.
3. The heterozygous variants identified in this study are proposed to block DRP1-dependent mitochondrial fission in cultured cells to an extent similar to the dominant negative K38A mutation that has been extensively studied (maybe with the exception of the mutation in P4, which reduces protein stability and thus leads to haploinsufficiency?). The only evidence shown in the study for this block is the indirect evidence of mitochondrial elongation (which as mentioned above is only in a subset of patient cell lines). It would be very convincing to see that these DRP1 variants inhibit stress-induced fragmentation induced by chemical or genetic approaches. This could be achieved by treating human fibroblasts with CCCP or siRNAs for MFNs or OPA1 or by ectopically expressing these variants in control cells, which should have the same effect.
4. Patient cells appear to have less intense TMRE staining than control cells, which would be consistent with an OXPHOS defect reported the western blot of Figure EV4 - DNM1L patient fibroblasts present with an OXPHOS defect. Western blots were reportedly performed in triplicate and these data should be quantified to support the statement that OXPHOS complexes are affected. It would be helpful to measure oxygen consumption rates in fibroblasts, especially given the incongruent observations

of normal respiratory chain measurements were made in muscle biopsies (P4) despite decreased COX2 levels (fibroblasts). These differences should be discussed since it is unclear how the OXPHOS observations made in patient muscle correlate (or not) with those made in patient fibroblasts. It was totally unclear to me whether OXPHOS defects are at all relevant in DRP1 deficiency.

5. SDS-PAGE of DRP1 levels in P4 show a decrease, consistent with haploinsufficiency but then in EV4A, but then in EV6 - BMH, which is presumably the same type of experiment, there seems to be even a bit more monomer DRP1 (top left) and then no difference for the right panel? This is very bizarre. How can this be explained? WB quantification of these experiments is required.

6. DRP1 recruitment studies in cultured cells are required to determine whether mutations affecting oligomerization or GTPase activity impact DRP1 recruitment to mitochondria (or peroxisomes) and subsequent division. These studies will also help resolve the puzzling observation that in vitro GTPase activity increase in some mutants that are associated with impaired mitochondrial morphology.

Minor comments:

- 1) It is regretful that none of the experiments are performed on the cells of patient 5. Although this patient harbors the same mutation as patient 1, the clinical manifestations are very different. The authors do not comment on that point. I imagine that no fibroblasts could be obtained from this patient?
- 2) In the figure 3B, it would be interesting to add a quantification in order to show in how many percent of cells those enlarged mitochondrial nucleoids are present and compare this percentage to control cells.
- 3) Figure 4: Showing V_{max} and k_{cat} is redundant, as it does not bring any supplementary information since they are proportional. k_{cat} is sufficient, as the authors do not comment on V_{max} . There is a typo in the y-axis of $k_{cat}/K_{0.5}$ (should be 0.25 and not 2.5)
- 4) The abstract sentence that reads "In vitro recombinant DRP1 protein harbouring patient variants demonstrated distinct effects on DRP1 protein stability, GTPase activity and oligomerisation and in vivo studies showed mitochondrial hyperfusion." -In the main "Disturbed organisation of mitochondria on a longitudinal plane was suggestive of mitochondrial hyperfusion" One TEM image of a single patient with no control image does not really correspond to in vivo studies in my opinion.
- 5) In figure EV1, there are no images of healthy controls. I imagine this is for ethical reasons (although it was possible to obtain fibroblasts from healthy controls). Can someone in the field just see by the look of A and B that there is increased lipid content and by the look of C and D the the organization of mitochondria is disturbed, suggestion hyperfusion? Because for me it is not clear.
- 6) In figure EV3: They show the STDEV only for the control muscle and control fibroblast. What is it the STDEV of? Repeated measures? Several controls?
- 7) I would be curious to know whether it would be possible to harness the predictive capabilities of AlphaFold2 to add further insights into the impact of the identified pathogenic variants.

Reviewer #2 (Comments to the Authors (Required)):

This paper reports the identification of four new mutations in DRP1 in 5 patients, and then illustrates the results of a characterization mainly based on biocomputational analysis but also on some experimental in vivo evidence based on the replication of equivalent mutations in DNM1 of yeast, which is considered the orthologue of human DRP1. The analysis is very detailed and rigorous, and the conclusion of the biocomputational structural investigation is that different mutations are associated with different, sometimes surprisingly different, molecular mechanisms of DRP1 malfunction. For instance two mutations seem to have opposite effects on the GTPase activity associated with DRP1, one increasing the other decreasing such activity. A third mutation is suggested to uncouple the GTPase activity from assembly, meaning I guess polymerization of the protein that is recruited to mitochondria from cytoplasm and forms ring-like structures that constrict the outer membrane of the organelle eventually inducing its split. This analysis is very painstaking and potentially interesting for a structural biologist but I sincerely doubt that it may be of any interest for a clinically oriented or even translationally medically oriented readership. The conclusion that different mutations act in different ways is almost self-evident and does not add any relevant information to understand the mechanism of disease. The suggestion of a possible segregation between some mutations and cardiac impairment is not powerful enough to be conclusive or even acceptably plausible, in the absence of a mechanistic explanation. I am a little surprised that other potentially relevant possibilities, including, for instance, the interaction of wt vs mutant DRP1 species with the mitochondrial partners performing the crucial role of recruiting the protein from the cytoplasm to the outer mitochondrial membrane have not been taken into consideration. Another missing part for a non-expert like me is the fact that one of the mutations could not be replicated in yeast because the resulting protein was impossible to be expressed; this would have prompted me to evaluate the levels of expression of each of the mutant protein and their actual amount in available tissues of patients. Despite the careful bioinformatic analysis reported in the paper, for instance concerning the different effects on the GTPase activity, the possible pathogenic mechanisms underpinning, in both cases, DRP1 impairment is proposed but not experimentally tested. In conclusion, the paper, although very complex and analytical, is not providing useful translational information which is the very dark side of the DRP1 mutations affecting humans, and the conclusions are in most cases descriptive rather than mechanistic.

Reviewer #3 (Comments to the Authors (Required)):

1. Meacham et al. describe four novel variants in DRP1 and delineate the divergent mechanisms by which they impact on DRP1 and mitochondrial dynamics. Many human DRP1 variants have previously been described, as has their impact on DRP1 and mitochondrial dynamics. Therefore, this work adds to this already existing body of knowledge, and provides some novel insight into how DRP1 assembly might be sensed by DRP1's GTPase domain.

The work is important because although work in the field of medical genetics has identified hundreds of new disease loci for mitochondrial disease in recent years, much is not known about the molecular and cellular impact of newly identified disease associated variants. This is an extremely thorough body of work that includes a reasonably detailed level of clinical data and fairly extensive functional characterisation of the four variants. The experimental data is of high quality and mostly supportive of the claims made. The manuscript is well written throughout. The authors also include a table including relevant information for all reported individuals with Drp1 variants which is helpful.

2. I do have a few issues or comments:

I do not understand why the authors claim that the L230dup fibroblasts have a hyperfused phenotype (pg 11 and fig 3), which is clearly not the case as shown by their MiNA analysis (Fig 3A). It is also stated in the supplementary table and discussion.

I thought that the evidence presented for the nucleoid/bulb structures presented in Fig 3B was not convincing. The lack of an image from control fibroblasts to compare with was not appropriate, given that in the control fibroblast images from Fig 3A similar bulb structures could be seen. I suggest performing an analysis that could provide quantitative data and statistical analysis to support this claim.

3.

A more detailed methodological description of the quadruple immunofluorescent assay is required, or if the original reference describing the assay is cited then at least an explanation should be provided why this is a "quadruple" assay given that only three proteins are mentioned - COXI, NDUFB8 and porin.

In the text and in the figure annotations it would be useful to always (or more often than has been done) include both Individual/Patient number and the corresponding DRP1 variant. Most relevant would be figure 3, Fig EV3, Fig EV4, EV6

The GTPase effector domain (GED) is not indicated in figure 1B, perhaps it could be highlighted?

In clinical report of P4 has a healthy younger sibling, not older, according to the pedigree?

Review 1

Reviewer #1 (Comments to the Authors (Required)):

The authors report an interesting analysis of four newly identified putatively pathogenic variants in *DNM1L* in five unrelated patients suffering from early-onset multi-systemic mitochondrial disease and make note of a peculiar manifestation of cardiac pathology, which is rather uncommon in patients carrying mutations in this gene. This synergistic study between structural biologists (US) and mitochondrial geneticists (UK) has the potential to be of great value for increasing our understanding of DRP1 biology in health and disease. They present extensive clinical background, as well as the data obtained from whole genome sequencing to identify heterozygous mutations that compromise DRP1 function, notably identifying the first patient with a pathogenic variant in the hinge region of the protein. They use *in silico* modeling of the different variants in order to understand the effect of those nucleic acid change on the protein 3D structure and function. Using recombinant proteins carrying most of these variants, they assess the impact these variants in DRP1 have on its oligomerization capacity and GTPase activity. Finally, they assess mitochondrial morphology, mtDNA nucleoid organization, and OXPHOS protein levels in primary fibroblasts from these patients. Their results bring new insights on the role of single nucleotide variants in mitochondrial dynamic and extend the list of disease-causing variants in this gene (which were nicely summarized in the supplemental table). The genetic, genomic, and structural data in particular are particularly convincing given their quantitative nature and also well presented, leaving little room for criticism (although I am not an expert in structural biology) and the interpretations of their data are, for the most part, parsimonious. Nevertheless, there are several points to address regarding the experiments performed on human cells carrying so-called pathogenic variants that must be addressed to satisfy the claims made in their study before this study is suitable for publication. These studies are essential to prove the pathogenic nature of these variants and may help clarify the surprising findings made *in vitro*. It is important that they be performed in quantitative fashion, which is lacking in all experiments (main and supplemental figures) performed on cultured cells with the exception of Figure 1A.

We thank reviewer 1 for their thoughtful comments and appreciate their positive remarks that this work "has the potential to be of great value for increasing our understanding of DRP1 biology in health and disease." We also appreciate the reviewer's recognition that the combination of genetic, genomic and structural data are "particularly convincing" and "well presented". We are grateful for their suggestions on how to improve the manuscript, particularly regarding their comments on cellular studies, and have provided specific answers and responses to each of their suggestions/concerns below in detail.

Major comments:

1. The authors state that "As expected, all four patient cell lines (P1-P4) demonstrated hyperfusion of mitochondrial networks relative to control cells, which was generally supported by quantification". However, in Figure 3A, for P3, the quantification shows no difference at all with C3 and C4, leading the reader to question this conclusion.
 - a. Is this because the experiment was only performed twice with only >20 fields of view? Why so few cells?

Thank you for noting the need for clarification in this conclusion. We have changed the manuscript to reflect that there is no marked hyperfusion to P3 cells when assessed using live imaging approaches (TMRM) Figure 3A-B. Upon visual inspection, the networks of Patient 3 (P3) stained with TMRM appeared hyperfused compared to adult controls despite there being no statistically significant differences in mean mitochondrial network length when using the Mitochondrial Analysis (MiNa) tool (FIJI/ImageJ). To further support these data, we performed additional analysis of the mitochondrial network length using immunofluorescence analysis of fixed *DNM1L*

patient (P1-P4) and age-matched control (C1, C2) fibroblasts labelled with TOM20 antibodies and analysed using the Columbus (PerkinElmer) software system, where at least 5500 mitochondria were analysed for each case. Consistently, P3 showed the lowest mean mitochondrial network length, however this was significantly higher than the control line. We are now including the additional data in a new Figure 3 and updated the result section (p10-11):

*“Analysis of mitochondrial networks using the ImageJ tool Mitochondrial Network Analysis (MiNA) revealed marked hyperfusion of mitochondria in P1, P2 and P4 compared to age-matched controls (Fig 3A-B). In addition, the mitochondrial network length was analysed using immunofluorescence labelling of fixed patient and age-matched control fibroblasts using TOM20 antibodies. The Columbus (Perkin Elmer) software system was used to quantify the hyperfusion of patient mitochondrial networks relative to controls. A minimum of 5,500 mitochondria were analysed for each case. Largely consistent with live cell imaging, significant hyperfusion of mitochondrial networks were observed in all four studied patient fibroblasts using this approach (Fig 3C). Whereas live cell imaging did not reveal extensive mitochondrial hyperfusion in P3 fibroblasts, TOM20 immunostaining revealed elongated mitochondria in P3 (p.Leu230dup) cells. Notably, these cells were the least affected compared to those from other patients (Fig 3C)...
...Although the degree of mitochondrial hyperfusion differed between patient fibroblasts, with P3 (p.Leu230dup) not displaying significant elongation by MiNA, this phenotype was consistent with previously reported de novo heterozygous DNMT1L variants (c.95G>C, p.Gly32Ala; c.436G>A, p.Asp146Asn; c.1184C>A, p.Ala395Asp; c.1207C>T, p.Arg403Cys; c.1292G>A, p.Cys431Tyr) and a GTPase deficient recombinant mutant (p.Lys38Ala) (Whitley et al, 2018; Longo et al, 2020; Chang et al, 2010; Waterham et al, 2007; Zhu et al, 2004).”*

In addition, we included minor changes in the discussion (p17), updated the materials and methods (p25-26) and provided a new Figure legend 3 (p42) to describe the data.

- b. It appears as though some cells in Figure 3 are in the process of mitosis, which is already known to modulate mitochondrial morphology. Have these cells been synchronized? Cell cycle is proven to modulate mitochondrial morphology and this point should be addressed. At the very least, they should be excluded from the analysis and not used as representative images.

Reviewer one is correct as the cells used for analysis in the original Figure 3 had not been synchronized to account for any differences in cell cycle stage. The decision to include these cells was based on other manuscripts such as Vanstone et al, 2016, EJHG; Whitley et al. 2018 HMG; Longo et al, 2020 HMG; Robertson et al., 2022 BioRxiv that included asynchronized cells in their network morphology analyses.

Indeed, the cell cycle can modulate network morphology wherein one would typically expect to see more interconnected networks during the G1 and G2 phases, and more fragmented networks during S-phase and mitosis (Horbay and Bilyy, 2016; Figure S1). Given DRP1 is critical for many of these cell cycle transitions, one might expect that the protein variants discussed in this manuscript may alter these transitions, perhaps further contributing to their mechanism of pathology. However, evaluating this is outside of the scope of this work, but something that needed to be mentioned in the manuscript (p4):

“Besides regulating mitochondrial metabolism, mitochondrial fission and fusion events play an essential role in a number of cellular processes, including cell cycle regulation (Horbay and Bilyy, 2016, Quian et al, 2012, Pangou and Sumara, 2021), immune response (Cervantes-Silva et al, 2021) and cell death (Aouacheria et al, 2017).”

Furthermore, we have removed any representative images of cells undergoing mitosis and synchronised the cells when generating the additional data assessing mitochondrial network length, peroxisomal morphology, DRP1 co-localisation and mtDNA nucleoids. In the methods section we now indicate that the mitochondrial network analysis in Figure 3A-B has been performed on asynchronised cells (p25): “*Asynchronised control and patient fibroblasts were cultured overnight on 35 mm Ibidi m-dishes...*”, whilst the data in Figure 3C-D and Figure 4 used synchronised cell lines (p25): “*Cells were synchronised overnight by starvation using DMEM media with 0.1% FBS (Mitra et al, 2009). G0-arrested cells were plated out on to 96-well plates and cultured in DMEM containing 10% FBS for 24 hours before incubation with 0.15% PicoGreen (Invitrogen P7581) for 30 mins at 37 °C...*” and “*For immunofluorescence analysis cells were synchronised as described above and fixed with 4% paraformaldehyde...*”

2. If the purported pathogenic variants negatively impact DRP1 function, cells carrying these variants should have blocked mitochondrial fission and blocked peroxisomal fission. The latter should be measured, especially since the original study describing a severe clinical outcome caused by DRP1 deficiency (Waterham 2007) exhibited both peroxisomal and mitochondrial morphology defects in fibroblasts.

The reviewers are well-motivated in their request for cell-based studies to evaluate the effects of these DRP1 pathological variants on peroxisomal morphology, DRP1 localization and/or potential dominant-negative behaviour of the variants in addition to the mitochondrial morphology analysis that was presented in the manuscript submitted.

We performed a more detailed characterisation of *DNM1L* patient fibroblasts to provide further evidence supporting the pathogenic effects of the studied variants. As requested, we examined the peroxisomal network in control and patient fibroblasts using immunofluorescence analysis of cells labelled with an antibody against a peroxisomal membrane marker PMP70. Quantitative image analysis of patient fibroblasts determined that the peroxisome length was elongated in all studied *DNM1L* patients (P1-P4) compared to controls (Figure 4A). Further, we performed a co-localisation analysis using the Pearson correlation coefficient (DRP1 vs. PMP70) and determined there was significantly decreased DRP1 co-localized with PMP70 in P1 and P2.

We updated the Materials and methods (p25), Figure legend 4 (p43), results (p11) and discussion (p17, p19-20) accordingly:

Results (p11):

“Given DRP1 has been implicated in both mitochondrial and peroxisomal fission (Waterham et al, 2007), we examined the effect of these variants on peroxisomal networks. Immunofluorescence labelling of control and DNM1L patient fibroblasts with antibodies against the peroxisomal membrane protein marker PMP70 was used to determine the peroxisomal morphology. The analysis using the Columbus software revealed that peroxisomes in P1 (p.Gly401Ser), P2 (p.Gly363Asp), P3 (p.Leu230dup) and P4 (p.Arg710Gly) appeared more fused with fewer overall numbers of peroxisomes and decreased size distribution, indicative of impaired fission (Fig 4A).

Co-localisation analysis between DRP1 and PMP70 showed decreased DRP1 at the peroxisomes in P1 and P2, but not P3 and P4, suggesting that the elongated peroxisomes in P4 are not simply due to decreased DRP1 recruitment (Fig 4B). Previous reports argue that not all DNM1L variants impair peroxisomal morphology, with several other variants in the GTPase domain having no impact on peroxisomal morphology despite affecting mitochondrial network morphology”

Discussion (p17) and (p19-20):

“Furthermore, DNM1L variants present in P1 (p.Gly401Ser), P2 (p.Gly363Asp), P3 (p.Leu230dup) and P4 (p.Arg710Gly) also impaired normal peroxisomal fission, which is not surprising given DRP1’s prominent role in this process (Tanaka et al, 2006; Kobayashi et al, 2007; Otera et al, 2010; Yamano et al, 2014; Koch & Brocard, 2012; Koch et al, 2005; Li & Gould, 2003; Gandre-Babbe & Van Der Bliiek, 2008). P1 (p.Gly401Ser), P2 (p.Gly363Asp) and P4 (p.Arg710Gly) DNM1L variants caused decreased DRP1 recruitment to the mitochondria (Fig 3D), but only P1 and P2 had decreased DRP1-peroxisome co-localisation (Fig 4B), suggesting that impaired DRP1 p.Arg710Gly peroxisomal fission occurs through a different mechanism. These data indicate that p.Arg710Gly mediated impairments are not simply due to a lack of DRP1 at the peroxisomal membrane, but may be due to impaired enzyme function with preservation of DRP1-peroxisome recruiter interactions, which are lost with the p.Gly363Asp and p.Gly401Ser variants.”

“Of note, P3 (p.Leu230dup) and P4 (p.Arg710Gly) also exhibited less severe peroxisomal defects compared to P1 (p.Gly401Ser) and P2 (p.Gly363Asp). It may be that concurrent mitochondrial and peroxisomal defects lead to more severe phenotypes and disease progression. Consistent with this, several other non-lethal DRP1 variants, located primarily in the GTPase domain, resulted in cells with normal peroxisome morphology despite having impaired mitochondrial networks (Gerber et al, 2017; Whitley et al, 2018; Chao et al, 2016) (Table S1).

*In true peroxisomal biogenesis disorders (PBDs), lipid metabolism, among other peroxisome-related metabolic pathways, is impaired. Clinically, DNM1L and PBD patients have phenotypic overlap including developmental delays, seizures, hypotonia, facial dysmorphism and vision impairment. Unlike PBD patients though, DNM1L patients do not typically develop renal or hepatic dysfunction, skeletal abnormalities, or cataracts (Waterham & Ebberink, 2012). Given these similarities, and the peroxisome fission abnormalities in many DNM1L patients, one might hypothesize that DNM1L patients would display similar biochemical profiles, with elevated very long-chain and branched-chain fatty acids (De Biase et al, 2019). Unfortunately, there remains a dearth of DNM1L patient reports that analyse both peroxisomal morphology and perform the necessary analyses to fully evaluate peroxisomal function. Based on data currently available, there is not a clear correlation between laboratory findings, peroxisome morphology, and disease severity with some variants displaying normal peroxisome morphology with normal laboratory tests p.Gly362Ser (Sheffer et al, 2016), normal peroxisomes with elevated plasma VLCFA and normal pristanic acid (p.Gly32Ala) (Whitley et al, 2018), abnormal peroxisomes with normal laboratory tests (p.Ser36Gly, p.Glu116Lysfs*6; p.Gly362Ser; p.Ile512Thr, p.Gly362Asp; p.Gly350Arg and p.Tyr691Cys) (Nasca et al, 2016; Verrigni et al, 2019; Chao et al, 2016), and abnormal peroxisomes with abnormal lab tests (p.Ala395Asp) (Waterham et al, 2007). Several studies noted abnormal peroxisomal morphology but did not perform lipid profiling (Longo et al, 2020; Chao et al, 2016; Zaha et al, 2016) and it is unclear whether these patients may have had abnormal results (Table S1). Although traditional peroxisome functional tests may not be fruitful diagnostically, future studies using lipidomic approaches may capture more nuanced metabolic changes that occur, identifying potential biomarkers for DNM1L-associated disease with peroxisome involvement. Ultimately, DNM1L disorders appear to derive primarily from mitochondrial defects and the degree of peroxisome-driven pathology remains unclear, but likely secondary.”*

Furthermore, we updated our Supplementary Table S1 describing the studies where DNM1L variants also affect peroxisomal morphology which now reads in the text as (p11-12):

“Specifically, the p.Glu2Ala, p.Ala192Glu, (Gerber et al, 2017), and p.Gly32Ala (Whitley et al, 2018) variants had normal peroxisomes in the setting of abnormal mitochondrial networks. Conversely, patient fibroblasts from a bi-allelic heterozygous patient carrying p.Ser36Gly;

*p.Glu116Lysfs*6 variants had both abnormal peroxisomal and mitochondrial fission (Nasca et al, 2016). Similar impairments were also observed in the p.Asp146Asn (Longo et al, 2020) and p.Gly223Val variants (Verrigni et al, 2019). (Table S1)."*

3. The heterozygous variants identified in this study are proposed to block DRP1-dependent mitochondrial fission in cultured cells to an extent similar to the dominant negative K38A mutation that has been extensively studied (maybe with the exception of the mutation in P4, which reduces protein stability and thus leads to haploinsufficiency?). The only evidence shown in the study for this block is the indirect evidence of mitochondrial elongation (which as mentioned above is only in a subset of patient cell lines). It would be very convincing to see that these DRP1 variants inhibit stress-induced fragmentation induced by chemical or genetic approaches. This could be achieved by treating human fibroblasts with CCCP or siRNAs for MFNs or OPA1 or by ectopically expressing these variants in control cells, which should have the same effect.

Reviewer one is correct that P4 (p.Arg710Gly) was found to have decreased levels of DRP1 by Western Blot (Figure S4A), as well as decreased protein stability as determined by thermal melt (Figure 6C-D). We appreciate the author's suggestion to evaluate whether the DRP1 variants could inhibit either stress-induced or even basal fission (expressed in the wild type background). We attempted to evaluate this, as well as determine whether these variants act in a dominant negative manner like many of the other previously reported DRP1 variants (Supplementary Table 1). We generated C-terminus GFP-tagged DRP1 wild type, patient DRP1 mutants (P1-P4) and the above mentioned K38A DRP1 mutant constructs (using the pcDNA3.1 vector), with a view to ectopically express these in wild type fibroblasts to assess the mitochondrial network and DRP1 localisation via microscopy.

Our initial attempts to transfect these constructs into both immortalised and primary control fibroblast cell lines led to poor transfection efficiency despite several weeks of troubleshooting using different ratios of DNA to transfection reagent, different brands of transfection reagents, and ultimately several different cell lines. Fortunately, we were able to attain 40-60% transfection rates in the U2OS cell line after 24 hour transfection, allowing for immunostaining of TOM20 and subsequent visualization of mitochondrial networks and DRP1::GFP.

Following image acquisition (please see Appendix 1), the expression of wild type DRP1::GFP appeared largely in two forms: i) as a diffused punctate pattern spread through the cytosol and mainly present in cells that have taken up less DRP1::GFP and the second form that appeared as ii) large bright aggregates. This pattern was similar in all studied DRP1 patient mutant forms with P1 presenting the least of the aggregated DRP1::GFP signal. The K38A DRP1::GFP mutant was particularly affected forming large GFP aggregates in the majority of cells.

Cells expressing the different forms of DRP1::GFP have been labelled with TOM20 antibody to visualise the mitochondrial network. The mitochondrial morphology in the majority of DRP1::GFP transfected cells appeared to be abnormal. As indicated in Appendix 1, we observed mitochondrial clustering/aggregating particularly around the perinuclear region as indicated by white arrows in both, wild type DRP1::GFP and DRP1::GFP mutant (P1-P4, K38A) transfected cells. In addition, some mitochondria formed long - what appeared to be hyperfused networks around the perinuclear region indicated by yellow arrows.

Although, there are some signs of mitochondrial hyperfusion when expressing the mutant forms of DRP1 in the studied cells, it is likely that the levels of DRP1::GFP plasmids introduced into the cells were toxic - hence the appearance of aggregated mitochondria. Therefore, we decreased

the expression levels by using differing lower amounts of DRP1 plasmid in combination with a filler pcDNA3.1- empty plasmid. Unfortunately, we could not detect any DRP1::GFP positive cells at these lower concentrations.

Although, it will be interesting to see the effect of these mutations on DRP1 localisation, mitochondrial network and stress-induced fragmentation, we believe this is something that can be addressed and further optimised in future studies using different DRP1 expression systems (e.g. mCherry or N-terminus tag). The effect on mitochondrial and peroxisomal network in *DNM1L* patients in combination with the DRP1 co-localisation studies and *in vitro* recombinant DRP1 work provide strong evidence that these *de novo* heterozygous variants are indeed disease causing.

4. Patient cells appear to have less intense TMRE staining than control cells, which would be consistent with an OXPHOS defect reported the western blot of Figure EV4 - *DNM1L* patient fibroblasts present with an OXPHOS defect. Western blots were reportedly performed in triplicate and these data should be quantified to support the statement that OXPHOS complexes are affected. It would be helpful to measure oxygen consumption rates in fibroblasts, especially given the incongruent observations of normal respiratory chain measurements were made in muscle biopsies (P4) despite decreased COX2 levels (fibroblasts). These differences should be discussed since it is unclear how the OXPHOS observations made in patient muscle correlate (or not) with those made in patient fibroblasts. It was totally unclear to me whether OXPHOS defects are at all relevant in DRP1 deficiency.

We regret the omission of Western Blot quantification throughout the manuscript. This is now included for all western blots in the manuscript (Figures S4 and S6) and details are highlighted in figure legends.

As it is expected that the multiple OXPHOS defects present in cells at a protein level usually affect the oxygen consumption (Bonnen et al, 2013 AJHG, Fig 4; Olahova et al, Brain, 2015, Figs 4&5; Alahmad et al, 2020, EMBO Mol Med, Figs 2&3), we did not feel that this would further the mechanistic understanding of how these protein variants result in pathology and believe the OXPHOS defect present in these cells is a secondary consequence of the disrupted mitochondrial network balance. We now added a sentence to the manuscript to clarify this (p14):

“Together these data suggest that different DNM1L variants have distinct impact on OXPHOS function in fibroblasts, with minimal correlations to disease onset or severity, suggesting the OXPHOS defects present in cells are a secondary consequence of the disrupted mitochondrial network balance as opposed to a driver of disease.”

We hypothesize that the differences between patient muscle and patient fibroblasts may arise from tissue specific effects. We have added a sentence to the manuscript to clarify this point so that it now reads (p14):

“Interestingly, there are some differences between the OXPHOS abnormalities between the patient muscle samples and fibroblasts. Most notably, P4 (p.Arg710Gly) whom had increased complex III activity in skeletal muscle, but decreased complex I and IV in fibroblasts. We hypothesize that these differences likely stem from tissue specific effects on respiration.”

SDS-PAGE of DRP1 levels in P4 show a decrease, consistent with haploinsufficiency but then in EV4A, but then in EV6 -BMH, which is presumably the same type of experiment, there seems to be even a bit more monomer DRP1 (top left) and then no difference for the right panel? This is very bizarre. How can this be explained? WB quantification of these experiments is required.

As mentioned above, we regret the omission of Western Blot quantification, and this has now been included for S4 and S6. The densitometric quantification of DRP1 relative to SDHA levels for S4A showed a mean 49% decrease in P4 (p.Arg710Gly) compared to adult control (Figure S4A, n = 3, STDEV = 13.1). The relative DRP1 band intensities detected in P1, P2 and P3 were similar to controls.

As we have indicated in the original figure legend S6: "*In A. and B. equal amounts of total cell lysates (50µg) were separated on a gradient (3-8%) Tris acetate gel.*" The experiment in A was performed once, as previously indicated in the figure legend and we regret that we cannot provide additional data showing equal loading of the samples. In A. we observed an increase in the amount of P4 oligomers, which was consistent with the *in vitro* SEC-MALS data where the R710G mutant (P4) was able to retain some ability to assemble into higher-order oligomeric species compared to G363D (P1) and G401S (P2) mutants (Figure 6A). Therefore, we focused on P4 and performed additional analysis of the DRP1 oligomeric status (Figure S6B).

We agree with the reviewer that it is not clear whether the amounts of monomeric DRP1 are decreased in P4 in Figure S6. To clarify this, we performed densitometric quantification of DRP1 in control and P4 samples (Figure S6B) and detected a mean 27% decrease (n = 3, STDEV = 2.3) and a 26% decrease (n = 2, STDEV = 7.5) in monomeric DRP1 levels in P4 relative to SDHA or HSP60, respectively. Together, the data in S4A and S6 suggest that DRP1 protein levels are decreased.

However, we would like to highlight the possible difference between these experiments that may have affected the observed DRP1 protein levels:

- i) Differences in sample preparation. Please note that samples in Figure S6 were treated in the presence of 1mM DMSO and a different lysis buffer (10 mM HEPES, 150 mM NaCl, 1 mM EGTA, 1% Triton-X (v/v), 1mM PMSF and 1x EDTA free protease inhibitor cocktail) at room temperature, following a 15 minutes treatment with 20mM DTT; whereas samples in Figure S4 were lysed on ice for 20 minutes in a lysis buffer containing 50 mM Tris-HCl pH 7.5, 130 mM NaCl, 2 mM MgCl₂, 1 mM PMSF, 1% Nonidet P-40 (v/v) and 1 x EDTA free protease inhibitor cocktail. We now include the lysis buffer composition in the supplementary information (p6).
- ii) The samples in Figure S4A were denatured by heat at 95°C for 5 minutes, which could potentially result in the formation of aggregates that cannot be fully resolved on 12% SDS PAGE; whereas the samples prepared for Figure S6 were heated at 70°C for 10 minutes in the presence of NuPAGE™ LDS Sample Buffer (ThermoFisher), which may be more representative of the actual amount of monomeric DRP1 protein present in P4 (p6).
- iii) The samples in Figure S6 were separated by a 3-8% Tris-Acetate gel to allow for the separation of larger oligomeric complexes, compared to the 12% SDS-PAGE used in Figure S4A.

In addition, we also commented on the above data in the discussion (p22):

"Supporting a direct disruption, R710G results in a lower T_m for the GTPase domain, albeit with retained nucleotide binding capabilities, reflective of decreased protein stability, possibly due to loss of the intramolecular GED-GTPase domain interactions. It is therefore not surprising that this patient had lower protein levels of DRP1 and this may be reflective of increased protein degradation secondary to the decreased stability; whereas the other patients did not, suggesting haploinsufficiency is not a major driver of pathology in those cases, which has been noted for other variants as well (Whitley et al, 2018)."

5. DRP1 recruitment studies in cultured cells are required to determine whether mutations affecting oligomerization or GTPase activity impact DRP1 recruitment to mitochondria (or peroxisomes) and subsequent division. These studies will also help resolve the puzzling observation that in vitro GTPase activity increase in some mutants that are associated with impaired mitochondrial morphology.

We agree with reviewer #1 that evaluating DRP1 recruitment would be beneficial in understanding how our *in vitro* results reconcile with in-cell processes. To evaluate this, we determined DRP1 co-localisation with mitochondria and peroxisomes in control and *DNM1L* patient (P1-P4) fibroblasts using immunofluorescence labelling with anti-DRP1, anti-TOM20 and anti-PMP70 antibodies (materials and methods have been updated accordingly: p25-26). Based on these studies, P1 (p.Gly401Ser), P2 (p.Gly363Asp), and P4 (p.Arg710Gly) demonstrated decreased DRP1 recruitment to the mitochondria (data shown in new Figure 3D) and described in the result section (p11)...

“To determine if mitochondrial network alterations were due to decreased DRP1 recruitment, we performed a co-localisation analysis using the Pearson’s co-localisation coefficient between DRP1 and TOM20 which showed decreased DRP1 at the mitochondria in P1, P2, and P4 fibroblasts. Of these, P4 had the most severe recruitment defect with the lowest Pearson’s R value and DRP1 appearing primarily cytosolic without punctate structures; which were still seen in other variants albeit to a lesser extent than the control fibroblasts (Fig 3D).”

...suggesting impaired DRP1-recruiter interactions either due to the variants inducing a less favourable oligomerisation state or direct disruption of the interaction. Therefore, the decrease in mitochondrial fission of variants that, perhaps counterintuitively, increase GTP hydrolysis is not that surprising. The text in the discussion section has been updated to reflect this new conclusion and reads (p18):

“Consistent with this, both p.Gly363Asp and p.Gly401Ser have decreased DRP1 at the mitochondria as determined by DRP1-TOM20 co-localisation analysis (Fig 3D). Therefore, the decrease in mitochondrial fission despite increased GTP hydrolysis for both G363D and G401S likely stems from a lack of DRP1 recruitment and productive fission activity at the mitochondria.”

As described above, we have also determined DRP1 co-localisation with peroxisomes (Figure 4B) and expanded the results (p11-12) and discussion (p17) section accordingly:

*“Co-localisation analysis between DRP1 and PMP70 showed decreased DRP1 at the peroxisomes in P1 and P2, but not P3 and P4, suggesting that the elongated peroxisomes in P4 are not simply due to decreased DRP1 recruitment (Fig 4B). Previous reports argue that not all DNM1L variants impair peroxisomal morphology, with several other variants in the GTPase domain having no impact on peroxisomal morphology despite affecting mitochondrial network morphology. Specifically, the p.Glu2Ala, p.Ala192Glu, (Gerber et al, 2017), and p.Gly32Ala (Whitley et al, 2018) variants had normal peroxisomes in the setting of abnormal mitochondrial networks. Conversely, patient fibroblasts from a bi-allelic heterozygous patient carrying p.Ser36Gly; p.Glu116Lysfs*6 variants had both abnormal peroxisomal and mitochondrial fission (Nasca et al, 2016). Similar impairments were also observed in the p.Asp146Asn (Longo et al, 2020) and p.Gly223Val variants (Verrigni et al, 2019). (Table S1).”*

“P1 (p.Gly401Ser), P2 (p.Gly363Asp) and P4 (p.Arg710Gly) DNM1L variants caused decreased DRP1 recruitment to the mitochondria (Fig 3D), but only P1 and P2 had decreased DRP1-

peroxisome co-localisation (Fig 4B), suggesting that impaired DRP1 p.Arg710Gly peroxisomal fission occurs through a different mechanism. These data indicate that p.Arg710Gly mediated impairments are not simply due to a lack of DRP1 at the peroxisomal membrane, but may be due to impaired enzyme function with preservation of DRP1-peroxisome recruiter interactions, which are lost with the p.Gly363Asp and p.Gly401Ser variants.”

Minor comments:

1. It is regretful that none of the experiments are performed on the cells of patient 5. Although this patient harbors the same mutation as patient 1, the clinical manifestations are very different. The authors do not comment on that point. I imagine that no fibroblasts could be obtained from this patient?

We agree with reviewer #1 that it is unfortunate that comparative experiments between P1 and P5 could not be performed, and they are correct that this may have been informative. However, a skin biopsy was never performed on this patient which prevented us from conducting these studies.

2. In the figure 3B, it would be interesting to add a quantification in order to show in how many percent of cells those enlarged mitochondrial nucleoids are present and compare this percentage to control cells.

The observation in the original Figure 3B is indeed interesting and something that has been previously reported in DRP1-defective cell lines (Ban-Ishihara *et al*, 2013; Ishihara *et al*, 2015; Ota *et al*. 2020). We have repeated the PicoGreen staining experiments on synchronised control and patient fibroblast cell lines, and following image capture, maximum projection images were analysed via the Columbus (PerkinElmer) software system to determine the proportion of enlarged nucleoids (area > 1.5 μm^2) in the total nucleoid pool, where the minimum of 3400 nucleoids were analysed in each subject. Although, enlarged nucleoids were present in all the analysed patient samples, significant differences were observed only in P1 and P2 compared to paediatric control (Figure S1C).

Increased levels of lipofuscin granules were detected in P4 fibroblasts during live imaging of PicoGreen stained cells. PicoGreen has an excitation peak at 480nm and an emission peak at 520nm, which overlaps with the lipofuscin autofluorescence signal (500-700nm). The increased levels of lipofuscin in P4 could be a sign of failed intracellular catabolism, leading to accumulation of damaged proteins and ultimately the accumulation of lipofuscin, comprising cell viability. Although, we observed enlarged mtDNA nucleoid structures in P4, we could not quantify the puncta accurately and therefore not included in the final analysis Figure S3D.

Furthermore, we now indicate that the passage number of all cell lines used in experimental studies was less than P0+12 from when they were received (p25):

“All primary control and patient fibroblasts used in this study were under P0+12 passages.”

3. Figure 4: Showing V_{max} and k_{cat} is redundant, as it does not bring any supplementary information since they are proportional. k_{cat} is sufficient, as the authors do not comment on V_{max} . There is a typo in the y-axis of $k_{cat}/K_{0.5}$ (should be 0.25 and not 2.5)

We have removed the V_{max} data from the figure and corrected the y-axis typo; we thank Reviewer #1 for highlighting this error.

4. The abstract sentence that reads "In vitro recombinant DRP1 protein harbouring patient variants demonstrated distinct effects on DRP1 protein stability, GTPase activity and oligomerisation and in vivo studies showed mitochondrial hyperfusion." -In the main "Disturbed organisation of mitochondria on a longitudinal plane was suggestive of mitochondrial hyperfusion" One TEM image of a single patient with no control image does not really correspond to in vivo studies in my opinion.

The conclusion of mitochondrial hyperfusion that reviewer #1 is referring to was based off the evaluation of mitochondrial network morphology in patient fibroblasts (Figure 3). We have now rephrased the sentence in the abstract to the following (p3):

"...in vitro recombinant human DRP1 mutants demonstrating greater impairments in protein oligomerisation, DRP1-peroxisomal recruitment and both mitochondrial and peroxisomal hyperfusion compared to GTPase or GTPase-effector domain variants".

As explained below (5.) we have now excluded the diagnostic TEM images from muscle of P1 (Original Figure S1) and only kept the information on the diagnostic TEM findings in the summary Table 1.

5. In figure EV1, there are no images of healthy controls. I imagine this is for ethical reasons (although it was possible to obtain fibroblasts from healthy controls). Can someone in the field just see by the look of A and B that there is increased lipid content and by the look of C and D the the organization of mitochondria is disturbed, suggestion hufefusion? Because for me it is not clear.

Reviewer #1 is correct and there are unfortunately no comparative images of healthy controls. Typically, these types of images would be obtained from unaffected siblings of the patient, or non-mitochondrial disease patients. However, muscle biopsy from healthy paediatric individuals is particularly difficult to get. The images in EV1 were **diagnostic** in nature and the only images that we could obtain from the referring diagnostic centre, thus our original decision to include them. However, we appreciate the reviewer informing us that the description of these images was not adequate for a non-expert and have removed them from the manuscript.

6. In figure EV3: They show the STDEV only for the control muscle and control fibroblast. What is it the STDEV of? Repeated measures? Several controls?

We thank reviewer #1 for highlighting this and have ensured this is now explained in the figure legend (p11) to show that the standard deviation shown in figure S3 is representative of the mean enzyme activities of control muscle cells (n=25) or fibroblasts (n=8) which have been scaled to 100% activity with the error bars representing the standard deviation of the activity.

7. I would be curious to know whether it would be possible to harness the predictive capabilities of Alphafold2 to add further insights into the impact of the identified pathogenic variants.

We thank reviewer #1 for this suggestion which prompted us to research this possibility further. Unfortunately, AlphaFold specifically states on their FAQ page that "AlphaFold has not been validated for predicting the effect of mutations. In particular, AlphaFold is not expected to produce an unfolded protein structure given a sequence containing a destabilising point mutation." (<https://alphafold.ebi.ac.uk/faq>)

In addition, a recent paper uploaded to BioRxiv found that output metrics from AlphaFold did not correlate well with changes in protein stability due to single point mutations (<https://www.biorxiv.org/content/10.1101/2021.09.19.460937v1.full.pdf>), further suggesting that AlphaFold was unfortunately not a reliable option for evaluating the structural impacts of these pathogenic variants.

Reviewer #2 (Comments to the Authors (Required)):

This paper reports the identification of four new mutations in DRP1 in 5 patients, and then illustrates the results of a characterization mainly based on biocomputational analysis but also on some experimental *in vivo* evidence based on the replication of equivalent mutations in DNM1 of yeast, which is considered the orthologue of human DRP1.

We apologize for any confusion we may have caused, but this paper utilized recombinant human DRP1 which was generated in *E. coli* and purified by affinity chromatography (Figures 5, 6 and S5), as well as cell-based studies utilizing patient-derived fibroblasts or muscle fibers. We opted not to use any yeast protein (DNM1) or yeast-based assays so that we could determine how the human proteins function as a result in their variations in amino acid sequence. To ensure clarity, we have gone back to the text and emphasized this such as “*we first expressed human DRP1 in recombinant form*” (p14) and “...with *in vitro* recombinant human DRP1 mutants demonstrating greater impairments in protein oligomerisation, DRP1-peroxisomal recruitment and both mitochondrial and peroxisomal hyperfusion compared to GTPase or GTPase-effector domain variants.” (p3). We have also ensured it is explicitly stated in the introduction that the human gene is referred to as *DNM1L* and the protein itself (such as in the *in vitro* assays using recombinant protein) is referred to as DRP1, consistent with standard naming convention in the field.

The analysis is very detailed and rigorous, and the conclusion of the biocomputational structural investigation is that different mutations are associated with different, sometimes surprisingly different, molecular mechanisms of DRP1 malfunction. For instance two mutations seem to have opposite effects on the GTPase activity associated with DRP1, one increasing the other decreasing such activity. A third mutation is suggested to uncouple the GTPase activity from assembly, meaning I guess polymerization of the protein that is recruited to mitochondria from cytoplasm and forms ring-like structures that constrict the outer membrane of the organelle eventually inducing its split.

Intramolecular interaction between a monomer's GTPase Effector Domain (GED), the N-terminal GTPase domain and stalk domain, as well as interactions between adjacent GEDs, are essential for regulation of DRP1 GTP hydrolysis (Pitts *et al*, 2004; Chang & Blackstone, 2007; Zhu *et al*, 2004). This is a common feature in all dynamin proteins (Zhang & Hinshaw, 2001; Muhlberg *et al*, 1997b; Smirnova *et al*, 1999; Sever *et al*, 1999b; Schumacher & Staeheli, 1998; di Paolo *et al*, 1999; Shin *et al*, 1999) where removal of the GED in dynamin or DRP1 does not prevent nucleotide binding or higher-order assembly but decreases GTPase activity (Muhlberg *et al*, 1997a; Zhu *et al*, 2004). Similarly, R710G can still bind GTP, evidenced by its ability to hydrolyze GTP and stabilization of the GTPase domain upon nucleotide binding, but has decreased GTPase activity. Mutation of R725 in dynamin (R710 in human DRP1 (NP_036192.2) and both located in the hinge 1 region) prevents stimulation of GTPase activity by the GED domain, suggesting it is a key residue involved in sensing and transmitting assembly information to the GTPase domain (Sever *et al*, 1999a).

Therefore, we predict that R710G is pathogenic due to a disruption in the sensing mechanism that facilitates assembly-driven increases in GTP hydrolysis. It is unclear if the substitution results in direct disruption of GED-GTPase domain interaction, or if it is a downstream mechanism.

This analysis is very painstaking and potentially interesting for a structural biologist but I sincerely doubt that it may be of any interest for a clinically oriented or even translationally medically oriented readership.

We respectfully disagree with Reviewer #2's opinion. Understanding the basis for protein dysfunction due to amino acid sequence changes is foundational for determining how and why protein variants can be pathological. Further, this determination can allow for the structure-based design of new therapeutic agents. Although not all protein variants may be amenable to drugging, single point substitutions that involve substitution of amino acids with larger, bulkier side chains for those with smaller side chains may be targetable by small molecules that can improve protein function. An example of this was reported by Hwang et al., in which a p.Arg459Leu mutation in the Glucose-6-phosphate-dehydrogenase protein (G6PD) was found to be a potent treatment for G6PD deficiency, hypothesized to be due to correction of structural defects caused by disrupted residue-residue interactions induced by this point mutation (Hwang et al., 2018. doi: 10.1038/s41467-018-06447-z). In our manuscript, we report a similar mutation of an Arginine to a small hydrophobic residue (glycine), which induces loss of a salt bridge interaction with the nearby glutamate at position 702 and resultant intramolecular instability. Like the G6PD mutation, it may be that certain point variants, such as DRP1 p.Arg710Gly, would be amenable to similar druggable approaches.

In addition, we hope that the reporting of four new pathological variants and the inclusion of a table summarizing all known reports of pathologic DRP1 variants, with associated patient pathology, laboratory and imaging studies, and cellular consequences will be of use to others in the field.

The conclusion that different mutations act in different ways is almost self-evident and does not add any relevant information to understand the mechanism of disease.

We were confused by this comment from reviewer #2. Perhaps the reviewer is referring to algorithms that can predict protein folds or stability from amino acid sequences (AlphaFold, Rosetta, Modeller, etc). However, few would characterise these programmes as self-evident and they currently cannot predict how single amino acid substitutions affect protein oligomerisation, hydrolytic activity, or cellular localization, all of which are central to DRP1 activity. Through the work presented in this manuscript, we experimentally determine that certain protein variants impact the protein's ability to localize to the mitochondria (p.Gly363Asp, p.Gly401Ser, and p.Arg710Gly) and peroxisome (p.Gly363Ser and p.Gly401Ser), self-assemble into higher order oligomers (p.Gly363Asp and p.Gly401Ser), and impair protein stability and likely the coupling of protein assembly with GTP hydrolysis (p.Arg710Gly). We hope this in-depth analysis will be an important resource to the field, where many previously published reports of patients with DRP1 variants focus more so on clinical presentation and whether or not mitochondrial or peroxisomal fission is disturbed without evaluating why it is altered.

With the push for personalized medicine, it will become increasingly important to understand how specific genetic mutations cause disease in patients. In this case, it is through differing mechanisms. Further, a thorough understanding of how individual mutations affect protein function is critical for furthering our ability to predict how and why certain protein variants may lead to disease, and what the manifestations of that disease may be.

The suggestion of a possible segregation between some mutations and cardiac impairment is not powerful enough to be conclusive or even acceptably plausible, in the absence of a mechanistic explanation.

We agree with reviewer #2 that the current correlations between specific DRP1 variants and cardiac impairment are limited, making diagnosis based solely on symptoms essentially impossible. However, from a clinical perspective, we think it is imperative to report on cardiac involvement in these patients so that other clinicians are aware this may be a presenting symptom associated with these rare disorders. As discussed in the manuscript, it is not necessarily surprising that DRP1 variants may lead to cardiac involvement as dilated cardiomyopathy was noted to be a complication in the python mouse model (C452F substitution in mouse DRP1 which is position p.Cys446 in human DRP1) Cahill et al, JBC 2015 (p19):

“Cardiac involvement in patients with DNM1L-related mitochondrial disease has previously been postulated, since a C452F substitution in mouse model DRP1 (position p.Cys446Phe in human DRP1 NP_036192.2) was shown to cause dilated cardiomyopathy (Cahill et al, 2015). Concordantly, a three-month old patient who initially presented with infantile parkinsonism like symptoms was identified to possess the same C446F substitution and died at 2.5 years of age due to sudden cardiac arrest (Díez et al, 2017). However, no post-mortem evaluation was performed to determine the cause of cardiac arrest.”

As reviewer #2 pointed out, given the current lack of understanding of why certain variants lead to cardiac involvement whereas others do not, we have revisited this paragraph in the discussion and worded our suggestions less strongly so that it now reads (p19):

“It would therefore seem appropriate that patients with confirmed pathogenic DNM1L variants follow a cardiac surveillance programme, as is in place for other forms of mitochondrial disease, with a view to appropriate pre-emptive treatment.”

I am a little surprised that other potentially relevant possibilities, including, for instance, the interaction of wt vs mutant DRP1 species with the mitochondrial partners performing the crucial role of recruiting the protein from the cytoplasm to the outer mitochondrial membrane have not been taken into consideration.

Reviewer #2 is correct that altered interactions between the DRP1 variants and recruiting proteins may be one reason for lack of DRP1 recruitment (now shown in Figure 4) and normal function, as determined by mitochondrial network morphology (Figure 3). We apologize for not making this point clearer in the manuscript. As submitted, the manuscript stated “DRP1 L230 is also near the DRP1-MID49 interface and the duplication may selectively inhibit recruiter interactions. Currently, only the structure of DRP1 in complex with MID49 has been solved (Kalia et al, 2018), so it is possible that other recruiting proteins bind at alternate locations enabling residual DRP1 activity to be performed” on page 18 lines 1-3 (original draft). Although this was in reference to why Leu230dup may have abnormal function, it is possible that the other substitution events (Gly401Ser, Gly363Asp, and Arg710Gly) could also lead to disruption of DRP1-recruiter interactions. Consistent with this, our new data suggest that p.Gly401Ser, p.Gly363Asp, and p.Arg710Gly had decreased DRP1-mitochondrial co-localisation, suggesting impaired recruitment. However, only p.Gly401Ser and p.Gly363Asp had impaired peroxisome recruitment, suggesting p.Arg710Gly may retain its ability to interact with a peroxisome-anchored recruiter and impaired peroxisome fission instead derives from impaired enzyme function (Please see Figure 3D and Figure 4B and updated results (p11-12) and discussion (p17-18 and p19 – highlighted in yellow) sections).

Given that only the structure of DRP1 bound to MID49 has been solved at this time, it is unclear where on DRP1 these other recruiting proteins, such as MFF and FIS1, may bind. This was alluded to on page 16 where we discussed a mutational hotspot within the stalk domain of the protein “a region important for mediating protein oligomerisation (Francy et al, 2017; Fröhlich et al, 2013b), which in turn is critical for stabilization of DRP1-MFF complexes post recruitment to the mitochondria (Clinton et al, 2016) as well as assembly with MID49 (Kalia et al, 2018).” We have expanded that section of the discussion as follows p18 to hopefully clarify this point:

“This suggests the variants may have impaired fission secondary to diminished higher-order assembly and/or poor recruitment to the mitochondria secondary to impaired DRP1-recruiter interactions”.

Another missing part for a non-expert like me is the fact that one of the mutations could not be replicated in yeast because the resulting protein was impossible to be expressed; this would have prompted me to evaluate the levels of expression of each of the mutant protein and their actual amount in available tissues of patients.

As described in the manuscript, we expressed human DRP1 in recombinant form using *E. coli* which was then purified for *in vitro* assays. Unfortunately, we were only able to obtain a truncated product of L230dup, as shown in the gel below and confirmed by LC-MS analysis. We have reworded the sentence in the manuscript text to make this clearer and it now reads (p14-15):

“Bacterial expression of all variants were similar to wild type (WT), except for L230dup which did not produce any full-length protein under multiple conditions and was unable to be purified for further studies.”

As reviewer #2 pointed out, it would be beneficial to understand if certain variants are simply less abundant than WT protein, which could lead to a perceived loss of function. Given this, we evaluated protein levels of DRP1 in the patient-derived fibroblasts that were available to us (shown in the original submission in Figure S4) which showed only P4 (p.Arg710Gly) had decreased levels of DRP1, which we concluded was likely due to decreased protein stability and was reflected in our *in vitro* studies as a decreased thermal melt temperature (Figure 6C-D). Additional tissue types were not banked, preventing a comparative analysis of protein expression in tissue types from being performed.

Despite the careful bioinformatic analysis reported in the paper, for instance concerning the different effects on the GTPase activity, the possible pathogenic mechanisms underpinning, in both cases, DRP1 impairment is proposed but not experimentally tested.

DRP1 is a large GTPase that functions to perform scission at the mitochondrial membrane. Per the current model in the field, this can only be accomplished when it is successfully recruited to

the mitochondria, assembles around the organelle, and hydrolyses GTP (Kalia et al, 2018; Fröhlich et al, 2013; Mears et al, 2011). If any of these steps are impaired, it can directly alter protein function, leading to pathological consequences. In this study, we extensively tested each variant for impaired function through analysis of protein assembly using size-exclusion chromatography with multi-angle laser light scattering (SEC-MALLS; Figure 6A-B), GTP hydrolysis via a regenerative coupled enzyme activity assay (Figure 5), and protein stability by thermal melt analysis of the protein alone and in the presence of nucleotide (Figure 6C-D).

Based on reviewer #2's, as well as the other reviewers' suggestions, we have now evaluated the effects of each protein variant on DRP1 localisation and also peroxisomal morphology, as DRP1 has been also involved in peroxisomal fission.

To provide further evidence supporting the pathogenic effects of the studied variants, we also examined the peroxisomal network in control and patient fibroblasts using immunofluorescence analysis of cells labelled with an antibody against a peroxisomal membrane marker PMP70. Quantitative image analysis of patient fibroblasts determined that the peroxisomal network morphology was elongated in all four studied patients compared to controls (Figure 4A). Furthermore, a co-localisation analysis was performed against DRP1 and the peroxisomal marker PMP70, which showed P1 and P2 had decreased DRP1 at the peroxisome (Figure 4B), whereas P4 had no change in DRP1-PMP70 co-localisation. We interpret these data to suggest that P4 p.Arg710Gly mediated impairments in peroxisomal fission are not simply due to a lack of DRP1 at the peroxisomal membrane, but likely are due to impaired enzyme function with preservation of DRP1-peroxisome recruiter interactions.

We updated the Materials and methods (p25), Figure legend 4 (p43), results (p11) and discussion (p17, p19-20) accordingly:

Results (p11):

"Given DRP1 has been implicated in both mitochondrial and peroxisomal fission (Waterham et al, 2007), we examined the effect of these variants on peroxisomal networks. Immunofluorescence labelling of control and DNM1L patient fibroblasts with antibodies against the peroxisomal membrane protein marker PMP70 was used to determine the peroxisomal morphology. The analysis using the Columbus software revealed that peroxisomes in P1 (p.Gly401Ser), P2 (p.Gly363Asp), P3 (p.Leu230dup) and P4 (p.Arg710Gly) appeared more fused with fewer overall numbers of peroxisomes and decreased size distribution, indicative of impaired fission (Fig 4A).

Co-localisation analysis between DRP1 and PMP70 showed decreased DRP1 at the peroxisomes in P1 and P2, but not P3 and P4, suggesting that the elongated peroxisomes in P4 are not simply due to decreased DRP1 recruitment (Fig 4B). Previous reports argue that not all DNM1L variants impair peroxisomal morphology, with several other variants in the GTPase domain having no impact on peroxisomal morphology despite affecting mitochondrial network morphology"

Discussion (p17) and (p19-20):

"Furthermore, DNM1L variants present in P1 (p.Gly401Ser), P2 (p.Gly363Asp), P3 (p.Leu230dup) and P4 (p.Arg710Gly) also impaired normal peroxisomal fission, which is not surprising given DRP1's prominent role in this process (Tanaka et al, 2006; Kobayashi et al, 2007; Otera et al, 2010; Yamano et al, 2014; Koch & Brocard, 2012; Koch et al, 2005; Li & Gould, 2003; Gandre-Babbe & Van Der Bliiek, 2008). P1 (p.Gly401Ser), P2 (p.Gly363Asp) and P4 (p.Arg710Gly) DNM1L variants caused decreased DRP1 recruitment to the mitochondria (Fig 3D), but only P1 and P2 had decreased DRP1-peroxisome co-localisation (Fig 4B), suggesting that impaired DRP1 p.Arg710Gly peroxisomal fission occurs through a different mechanism. These

data indicate that p.Arg710Gly mediated impairments are not simply due to a lack of DRP1 at the peroxisomal membrane, but may be due to impaired enzyme function with preservation of DRP1-peroxisome recruiter interactions, which are lost with the p.Gly363Asp and p.Gly401Ser variants.”

In addition we examined the co-localisation of DRP1 and mitochondria in control and patient fibroblasts which showed P1 (p.Gly401Ser), P2 (p.Gly363Asp), and P4 (p.Arg710Gly) had decreased DRP1 recruitment to the mitochondria (Figure 3D), suggesting impaired DRP1-recruiter interactions either due to the variants inducing a less favourable oligomerisation state or direct disruption of the interaction. Similar to the co-localisation analysis of DRP1 and peroxisomes, P3 did not have impaired DRP1 localization to the mitochondria. We have updated the methods (p25-26), results (p11) and discussion (p18) to reflect these new findings.

Results (p11):

“To determine if mitochondrial network alterations were due to decreased DRP1 recruitment, we performed a co-localisation analysis using the Pearson’s co-localisation coefficient between DRP1 and TOM20 which showed decreased DRP1 at the mitochondria in P1, P2, and P4 fibroblasts. Of these, P4 had the most severe recruitment defect with the lowest Pearson’s R value and DRP1 appearing primarily cytosolic without punctate structures; which were still seen in other variants albeit to a lesser extent than the control fibroblasts (Fig 3D).”

Discussion (p18):

“Consistent with this, both p.Gly363Asp and p.Gly401Ser have decreased DRP1 at the mitochondria as determined by DRP1-TOM20 co-localisation analysis (Fig 3D). Therefore, the decrease in mitochondrial fission despite increased GTP hydrolysis for both G363D and G401S likely stems from a lack of DRP1 recruitment and productive fission activity at the mitochondria.”

Reviewer #3 (Comments to the Authors (Required)):

Meacham et al. describe four novel variants in DRP1 and delineate the divergent mechanisms by which they impact on DRP1 and mitochondrial dynamics. Many human DRP1 variants have previously been described, as has their impact on DRP1 and mitochondrial dynamics. Therefore, this work adds to this already existing body of knowledge, and provides some novel insight into how DRP1 assembly might be sensed by DRP1's GTPase domain. The work is important because although work in the field of medical genetics has identified hundreds of new disease loci for mitochondrial disease in recent years, much is not known about the molecular and cellular impact of newly identified disease associated variants. This is an extremely thorough body of work that includes a reasonably detailed level of clinical data and fairly extensive functional characterisation of the four variants. The experimental data is of high quality and mostly supportive of the claims made. The manuscript is well written throughout. The authors also include a table including relevant information for all reported individuals with Drp1 variants which is helpful.

We appreciate reviewer #3's helpful comments on what they consider an “extremely thorough body of work” and agree that this work is important given the dearth of mechanistic understanding of how mitochondrial disease associated variants cause pathology. We also appreciate their recognition of the supplemental table that we generated and hope it will be a helpful resource for the field.

1. I do not understand why the authors claim that the L230dup fibroblasts have a hyperfused phenotype (pg 11 and fig 3), which is clearly not the case as shown by their MiNA analysis (Fig 3A). It is also stated in the supplementary table and discussion. I thought that the

evidence presented for the nucleoid/bulb structures presented in Fig 3B was not convincing. The lack of an image from control fibroblasts to compare with was not appropriate, given that in the control fibroblast images from Fig 3A similar bulb structures could be seen. I suggest performing an analysis that could provide quantitative data and statistical analysis to support this claim.

We thank reviewer #3 for noting this discrepancy in our conclusion regarding the network morphology of P3 (pL230dup) and apologize for any conclusion this may have caused. During image analysis, the networks of P3 appeared to be hyperfused compared to the control samples (new Figure 3A), despite there being no statistically significant differences in the mean mitochondrial network length by MiNA (new Figure 3B). To clarify this, we have updated the text and performed additional assessment of mitochondrial network length using immunofluorescence labelling of mitochondria with TOM20 antibodies and mitochondrial network length analysis performed on the Columbus (PerkinElmer) software system. We have included a new Figure 3 and legend (p42), additional methods (p25-26), and updated the results (p10-11) and discussion (p17) section accordingly.

Results (p10-11):

*“Analysis of mitochondrial networks using the ImageJ tool Mitochondrial Network Analysis (MiNA) revealed marked hyperfusion of mitochondria in P1, P2 and P4 compared to age-matched controls (Fig 3A-B). In addition, the mitochondrial network length was analysed using immunofluorescence labelling of fixed patient and age-matched control fibroblasts using TOM20 antibodies. The Columbus (Perkin Elmer) software system was used to quantify the hyperfusion of patient mitochondrial networks relative to controls. A minimum of 5,500 mitochondria were analysed for each case. Largely consistent with live cell imaging, significant hyperfusion of mitochondrial networks were observed in all four studied patient fibroblasts using this approach (Fig 3C). Whereas live cell imaging did not reveal extensive mitochondrial hyperfusion in P3 fibroblasts, TOM20 immunostaining revealed elongated mitochondria in P3 (p.Leu230dup) cells. Notably, these cells were the least affected compared to those from other patients (Fig 3C)...
...Although the degree of mitochondrial hyperfusion differed between patient fibroblasts, with P3 (p.Leu230dup) not displaying significant elongation by MiNA, this phenotype was consistent with previously reported de novo heterozygous DNMT1L variants (c.95G>C, p.Gly32Ala; c.436G>A, p.Asp146Asn; c.1184C>A, p.Ala395Asp; c.1207C>T, p.Arg403Cys; c.1292G>A, p.Cys431Tyr) and a GTPase deficient recombinant mutant (p.Lys38Ala) (Whitley et al, 2018; Longo et al, 2020; Chang et al, 2010; Waterham et al, 2007; Zhu et al, 2004).”*

The observation of enlarged nucleoids in the original Figure 3B is indeed interesting and something that has been previously reported in DRP1-defective cell lines (Ban-Ishihara et al, 2013; Ishihara et al, 2015; Ota et al. 2020). We have performed the PicoGreen staining experiments on control and *DNMT1L* patient (P1-P4) fibroblast cell lines, and following image capture, maximum projection images were analysed via the Columbus (PerkinElmer) software system to determine the proportion of enlarged nucleoids (area > 1.5µm²) in the total nucleoid pool, where the minimum of 3400 nucleoids were analysed in each subject. Although, enlarged nucleoids were present in all the analysed patient samples, significant differences were observed only in P1 and P2 compared to paediatric control (Figure S1C).

Increased levels of lipofuscin granules were detected in P4 fibroblasts during live imaging of PicoGreen stained cells. PicoGreen has an excitation peak at 480nm and an emission peak at 520nm, which overlaps with the lipofuscin autofluorescence signal (500-700nm). The increased levels of lipofuscin in P4 could be a sign of failed intracellular catabolism, leading to accumulation

of damaged proteins and ultimately the accumulation of lipofuscin, comprising cell viability. Although, we observed enlarged mtDNA nucleoid structures in P4, we could not quantify the puncta accurately and therefore not included in the final analysis Figure S3D.

We updated the results (p12), discussion (p17) and methods (p25) accordingly:

“Detailed analysis of mtDNA nucleoids stained with PicoGreen using Columbus software (PerkinElmer) revealed marked differences in the proportion of enlarged nucleoids (area > 1.5µm²) in P1 (p.Gly401Ser) and P2 (p.Gly363Asp) compared to control (Fig S1C). There was no difference in nucleoid size ratio between P3 (p.Leu230dup) and control (Fig S1C). Although, upon visual examination P4 (p.Arg710Gly) nucleoids appeared enlarged compare to controls, we were not able to accurately quantify the individual puncta due to increased levels of lipofuscin present in these cells (Fig S1D).”

“Analysis of mitochondrial network morphology in fixed patient-derived cell lines revealed impaired mitochondrial fission leading to hyperfused mitochondrial networks (Fig 3C) and in some cases enlarged mtDNA nucleoids (Fig S1), confirming dysfunctional DRP1 as the primary pathogenic factor in these patients.”

“Cells were synchronised overnight by starvation using DMEM media with 0.1% FBS (Mitra et al, 2009) G0-arrested cells were plated out on to 96-well plates and cultured in DMEM containing 10% FBS for 24 hours before incubation with 0.15% PicoGreen (Invitrogen P7581) for 30 mins at 37 °C. Following three washes in Fluorobrite DMEM (Fisher Scientific A1896701) Z-stack images were taken on a Zeiss CellDiscoverer7 microscope with 50x water objective (NA 1.2). Maximum projection images were analysed using the Columbus (PerkinElmer) software system and for each image field the proportion of enlarged nucleoids classed as over 1.5µm² were calculated in the total nucleoid pool. Statistically significant differences were calculated via non-parametric one-way ANOVA using GraphPad Prism.”

2. A more detailed methodological description of the quadruple immunofluorescent assay is required, or if the original reference describing the assay is cited then at least an explanation should be provided why this is a "quadruple" assay given that only three proteins are mentioned - COXI, NDUFB8 and porin.

We have expanded our description of the quadruple assay in the results (p13) and methods section so that it now reads (p23):

“Complex I and IV immunodeficient muscle fibres in P2 were determined by a quadruple fluorescent IHC assay of OXPHOS function, which evaluates protein levels of mitochondrial subunits of complex I (NDUFB8) and complex IV (COX1). In addition to the immunofluorescence labelling of muscle sections using antibodies against the above described OXPHOS complexes, the mitochondrial mass was quantified using an antibody against the outer mitochondrial membrane protein – porin (VDAC) and myofibre boundaries were labelled with anti-laminin, a membrane glycoprotein as previously described (Rocha et al, 2015).”

3. In the text and in the figure annotations it would be useful to always (or more often than has been done) include both Individual/Patient number and the corresponding DRP1 variant. Most relevant would be figure 3, Fig EV3, Fig EV4, EV6.

Thank you reviewer #3 for this suggestion, which we think has increased the clarity and readability of the manuscript. We have ensured all mentions of individual/patient numbers are now accompanied by their corresponding DRP1 variant in both figures and in the manuscript text.

4. The GTPase effector domain (GED) is not indicated in figure 1B, perhaps it could be highlighted?

The GTPase effector domain (GED) is now indicated in Figure 1B.

5. In clinical report of P4 has a healthy younger sibling, not older, according to the pedigree?

We thank reviewer #3 for his observation and double checked the clinical records for P4. As stated The Supplementary information states:

" Patient 4 (P4) was born to non-consanguineous parents and has a healthy older sibling."

This is correct, however we made a mistake in the pedigree in Figure 1B and indicated that P4 was the older sibling. This has now been amended.

Appendix 1

June 29, 2022

RE: Life Science Alliance Manuscript #LSA-2021-01284-TR

Dr. Monika Oláhová
Newcastle University
Wellcome Centre for Mitochondrial Research
Framlington Place
Newcastle upon Tyne NE2 4HH
United Kingdom

Dear Dr. Oláhová,

Thank you for submitting your revised manuscript entitled "Novel DNM1L variants impair mitochondrial dynamics through divergent mechanisms.". We would be happy to publish your paper in Life Science Alliance pending final revisions necessary to meet our formatting guidelines.

- please upload your supplementary figure files as single files
- please add a callout for Figure 5B-D to your main manuscript text
- in the Materials and Methods section, please rename the section entitled "R Scripts" to "Data Availability Statement"

Figure Check:

- Figure S1B, panel C2, it seems the boxed area could match the zoomed panel better
- are there any splices in the blots for Figure S4B? If so, please indicate with a vertical line and indicate in legend

A. FINAL FILES:

B. MANUSCRIPT ORGANIZATION AND FORMATTING:

Sincerely,

Reviewer #1 (Comments to the Authors (Required)):

The authors have provided convincing and compelling additional experimental evidence and explanations of their data to warrant publication. This study will be of value to the scientific and medical communities. I commend the authors for the thoughtful and careful quantitative cellular studies performed during the revision. Please be mindful of typos in the Figure S4B.

Reviewer #3 (Comments to the Authors (Required)):

The authors have addressed satisfactorily all of the comments I made on the first submission. This includes most importantly additional analyses related to Fig 3 as well as other minor comments on the manuscript.

July 7, 2022

RE: Life Science Alliance Manuscript #LSA-2021-01284-TRR

Dr. Monika Oláhová
Newcastle University
Wellcome Centre for Mitochondrial Research
Framlington Place
Newcastle upon Tyne NE2 4HH
United Kingdom

Dear Dr. Oláhová,

Thank you for submitting your Research Article entitled "Novel DNM1L variants impair mitochondrial dynamics through divergent mechanisms.". It is a pleasure to let you know that your manuscript is now accepted for publication in Life Science Alliance. Congratulations on this interesting work.

DISTRIBUTION OF MATERIALS:

Again, congratulations on a very nice paper. I hope you found the review process to be constructive and are pleased with how the manuscript was handled editorially. We look forward to future exciting submissions from your lab.

Sincerely,
